# A penicillin-binding protein inhibitor series to target drug-resistant *Neisseria gonorrhoeae*

Tsuyoshi Uehara [1] ✉, Allison L. Zulli[1,7], Brittany Miller[1,7], Lindsay M. Avery[1], Steven A. Boyd [1], Cassandra L. Chatwin[1], Guo-Hua Chu[1], Anthony S. Drager[1], Mitchell Edwards[1], Susan G. Emeigh Hart[1], Nathan J. Line[1], Cullen L. Myers[1], Gopinath Rongala [1], Annie Stevenson [1], Kyoko Uehara[1], Fan Yi [1], Bibo Wang[2], Zhenwu Liu[2], Mingyue Wang[2], Zhichao Zhao[2], Xinming Zhou[2], Haiyan Zhao[2], Caleb M. Stratton [3], Sandeepchowdary Bala[3], Christopher Davies[3], Rok Tkavc[4,5], Ann E. Jerse [5], Daniel C. Pevear[1,6], Christopher J. Burns[1], Denis M. Daigle[1] & Stephen M. Condon [1] ✉

Emerging multidrug-resistant *Neisseria gonorrhoeae* strains possessing altered *penA* alleles (encoding penicillin-binding protein 2, PBP2) threaten the utility of ceftriaxone, the last remaining outpatient antibiotic for gonorrhoea treatment, posing a global health emergency. Here we report a benzoxaborinine-based penicillin-binding protein inhibitor series (boro-PBPi) developed to address *penA*-mediated ceftriaxone resistance. Optimization of boro-PBPi led to the identification of compound 21 (VNRX-14079), which exhibited potent antibacterial activity against multidrug-resistant *N. gonorrhoeae* through high-affinity binding to the PBP2 target. Boro-PBPi–PBP2 complex structures confirmed the covalent interaction of the boron atom with the catalytic residue Ser310 and the importance of the $\beta_3$–$\beta_4$ loop mobility for improved affinity. Boro-PBPi 21 elicits bactericidal activity, a low frequency of resistance, a good safety profile, suitable pharmacokinetic properties and in vivo efficacy in a murine infection model against ceftriaxone-resistant *N. gonorrhoeae*. Boro-PBPi 21 is therefore a promising antigonorrhoea agent poised for further advancement.

Gonorrhoea, caused by *Neisseria gonorrhoeae*, is the second-most prevalent sexually transmitted bacterial infection, with an estimated 82.4 million cases worldwide in 2020 (ref. 1). If left untreated, gonorrhoea can lead to serious health complications, including infertility, pelvic inflammatory disease and increased risk of human immunodeficiency virus/AIDS[1]. Although gonorrhoea is treatable and can be cured by existing antibiotics, multidrug-resistant *N. gonorrhoeae* is emerging, making the treatment of gonorrhoea more challenging[2–7]. The current

last-line therapeutic option to treat uncomplicated gonorrhoea is a single intramuscular (IM) dose of an extended-spectrum cephalosporin, ceftriaxone (CRO)[8]. Unfortunately, global CRO-resistant *N. gonorrhoeae* infections are increasing in frequency, foreshadowing an era of untreatable gonorrhoea infections, which would result in an unsustainable burden on healthcare systems[9,10]. Because there are no effective vaccines, antibiotic therapy remains the only option for treatment[11]. Therefore, the Centers for Disease Control and Prevention (CDC) and the World

[1]Venatorx Pharmaceuticals, Inc., Malvern, PA, USA. [2]BioDuro-Sundia, Beijing, China. [3]Department of Biochemistry and Molecular Biology, University of South Alabama, Mobile, AL, USA. [4]Henry M. Jackson Foundation for the Advancement of Military Medicine, Bethesda, MD, USA. [5]Department of Microbiology and Immunology, Uniformed Services University of the Health Sciences, Bethesda, MD, USA. [6]Spring Mill Pharma Inc., Malvern, PA, USA. [7]These authors contributed equally: Allison L. Zulli, Brittany Miller. ✉e-mail: tsuyoshi.uehara@gmail.com; stephencondon3418@gmail.com

**Fig. 1 | Evolution of boro-PBPi from β-lactamase inhibitor-containing benzoxaborinine core to ureido-containing lead compound 21 (VNRX-14079).** Compound **1** was obtained by introducing the side chain present in CRO. The introduction of the side chain present in cefoperazone yielded compound **2**, which exhibited better activity than **1**. Based on the structure-activity relationship, negatively charged moieties were incorporated, yielding 12 (benzoate) and **15** (aryl phosphonate). Substitution of the dioxopiperidinyl moiety of **12** with the imidazolinone present in mezlocillin afforded compound **18**, which showed equivalent activity to **12**. Finally, the addition of multiple fluorine atoms to **18** yielded boro-PBPi **21**, which exhibited improved activity. The biochemical and anti-gonorrhoeal activities of each compound are shown in Table 1.

Health Organization (WHO) have prioritized the discovery and development of new antibiotics against multidrug-resistant *N. gonorrhoeae*[12–14].

Protein variants of *penA*, encoding penicillin-binding protein 2 (PBP2; functionally equivalent to the cell division protein PBP3 in *Escherichia coli*), are the predominant mechanism of CRO resistance in *N. gonorrhoeae*, with the most urgent threat mediated by the mosaic *penA-60* allele (a 'mosaic PBP2' refers to a variant of PBP2 protein composed of segments derived from several *Neisseria* species and are often associated with decreased susceptibility to cephalosporin antibiotics)[5,10]. *N. gonorrhoeae* strains possessing *penA-60* have been spreading globally since the discovery of this allele in 2015 in the FC428 strain isolated in Japan[9,15–20].

We have identified a series of non-β-lactam inhibitors of PBP transpeptidase function (PBPi), based on the α-alkylamido benzoxaborinine scaffold present in the β-lactamase inhibitors taniborbactam[21,22] and ledaborbactam etzadroxil[23,24] (Supplementary Fig. 1). Owing to the similarity within the β-lactamase and PBP active sites, boronate-based inhibitors can exploit several common binding elements, including the slowly reversible interaction of the active site serine hydroxyl group with the boron atom of the benzoxaborinine core[21,22,25–27]. Here we present a benzoxaborinine-based PBP inhibitor series (boro-PBPi) that exhibits activity against CRO-resistant strains of *N. gonorrhoeae* containing mosaic PBP2. Moreover, we provide in vitro and in vivo data to support further preclinical development of boro-PBPi **21** (VNRX-14079) as a potential new agent for the treatment of gonorrhoea, including those resistant to the current standard of care agent CRO.

## Results

### Identification of benzoxaborinine-based PBPi with stand-alone antibacterial activity

To adapt a boron-based warhead to inhibit PBPs like the β-lactam ring, we introduced the aminothiazole-containing oxyimino-acetamide side chain present in the third-generation cephalosporins cefdinir and CRO (Supplementary Fig. 2) onto the benzoxaborinine warhead, generating boro-PBPi **1** (Fig. 1). In vitro testing of compound **1** against a selected panel of Gram-negative organisms revealed modest single-agent activity against certain *E. coli* and *Klebsiella pneumoniae* strains (Extended Data Table 1). Notably, **1** exhibited only fourfold improved activity (minimal inhibitory concentration (MIC) of 8 µg ml⁻¹) against the hyperpermeable *E. coli* BAS901C strain relative to the American Type Culture Collection (ATCC) 25922 wild-type strain, suggesting that its antibacterial activity might be limited by its weak affinity for target PBPs (*E. coli* PBP3 half-maximum inhibitory concentration (IC$_{50}$) of 113 µM). Importantly, **1** exhibited modest activity against non-mosaic PBP2-producing *N. gonorrhoeae* ATCC 49226 (MIC of 16 µg ml⁻¹), with an IC$_{50}$ of 100 µM against wild-type PBP2 (PBP2$^{WT}$) isolated from the *N. gonorrhoeae* FA19 strain (Table 1). However, **1** was inactive (MIC of >128 µg ml⁻¹) against the H041 strain (also known as WHO X) producing mosaic PBP2 (encoded by *penA-37*) that mediates CRO resistance.

### Incorporation of the ureido moiety for improved activity against *N. gonorrhoeae*

Inspired by β-lactam structures, the oxyimino amide side chain was replaced with the ureido-based aryl glycine motif present in piperacillin and cefoperazone (Supplementary Fig. 2). The ureido-containing benzoxaborinines were initially prepared as near equimolar mixtures of the *R,R*- and *S,R*-diastereomers; when required, the desired *R,R*-diastereomers (*infra*) were isolated by reversed-phase high-performance liquid chromatography. The phenol-bearing boro-PBPi **2** (Fig. 1) showed moderate antibacterial activity against representative *N. gonorrhoeae* strains producing non-mosaic PBP2 and exhibited good in vitro binding to PBP2$^{WT}$ (IC$_{50}$ of 2.8 µM; Table 1). However, poor binding was observed for compound **2** against the mosaic PBP2 protein derived from the H041 strain (PBP2$^{H041}$; IC$_{50}$ of 770 µM). The MICs for **2** were correspondingly higher against H041 compared with ATCC 49226 with non-mosaic PBP2 (Table 1).

Using the 4-hydroxyphenyl glycine moiety present in compound **2** as a handle, a series of substituted phenol-bearing boro-PBPi was prepared. In both the CRO-susceptible ATCC 49226 and CRO-resistant H041 strains, the MICs of *R,R*-**3** were comparable to those of **2**, which was an approximately 1:1 mixture of active and inactive diastereomers. *R,R*-**4**, bearing a fluoro on the benzo ring, was less active against

**Table 1 | PBP2 binding and minimal inhibitory concentration data for ceftriaxone, cefoperazone and representative boro-PBPi**

| Compound | *N. gonorrhoeae* PBP2 binding mean $IC_{50}$ (µM) | | *N. gonorrhoeae*, non-mosaic PBP2 MIC (µg ml⁻¹) | | *N. gonorrhoeae*, mosaic PBP2 MIC (µg ml⁻¹) | | |
|---|---|---|---|---|---|---|---|
| | FA19, wild type | H041, mosaic | ATCC 49226 | FA1090 | WHO K | WHO Q (G7944) | H041 (WHO X) |
| CRO | 0.3 | <0.5 | 0.008 | 0.004 | 0.03 | 0.5 | 1 |
| Cefoperazone | 0.3 | <0.5 | 0.06 | ≤0.008 | 0.12 | 0.25 | 0.25 |
| **1** | 100 | 376 | 16 | 4 | 64 | 64 | >128 |
| **2** | 2.8 | 770 | 4 | 0.25 | 32 | 16 | 64 |
| **12** | 1.0 | 42 | 0.5 | 0.06 | 4 | 1 | 4 |
| **15** | 0.7 | 7.1 | 0.06 | 0.016 | 0.25 | 0.12 | 0.5 |
| **18** | 0.3 | 3.2 | 0.12 | 0.016 | 0.25 | 0.12 | 1 |
| **21** | 1.7 | 3.0 | 0.06 | 0.03 | 0.12 | 0.06 | 0.12 |

ATCC 49226 than **3** and exhibited only modest activity against the CRO-resistant H041 strain (MIC of 16 µg ml⁻¹; Extended Data Table 2). Additional fluorinated analogues **5**–**9** revealed improved PBP2$^{WT}$ binding relative to non-fluorinated **2** and had comparable or lower MIC values against ATCC 49226. Although most of these analogues plateaued at an MIC of 16–32 µg ml⁻¹ against CRO-resistant H041, trifluorinated **9** displayed a slightly improved MIC of 8 µg ml⁻¹ against H041, suggesting that the inclusion of fluoro groups may have a positive effect on PBP2 binding and/or periplasmic accumulation (Extended Data Table 2).

To understand the relationship between fluoro substitution and PBP2 binding and/or antibacterial activity, p$K_a$ values of the phenol proton for these analogues were calculated. Although unsubstituted phenol has a p$K_a$ (Ar-OH) of 10.0, the inductive effect of fluoro substitution lowers the p$K_a$ of 2-fluoro- and 3-fluoro-phenol to 8.7 and 9.3, respectively. The 2,3,6-trifluoro-4-hydroxyphenyl glycine of **9** was calculated to have a p$K_a$ of 6.7, suggesting that **9** probably exists as a phenoxide anion at physiological pH, which may contribute to the improved PBP2 binding and antibacterial activity.

### Benzoate- and phenyl phosphonate-containing benzoxaborinines improve boro-PBPi activity

Expecting that a carboxylate residue at the *para* position of the aryl glycine moiety might be preferred over a phenoxide anion, a series of negatively charged benzoate-containing boro-PBPi was synthesized. Boro-PBPi **10** and **11** displayed good activity against ATCC 49226 (MIC of 1 and 4 µg ml⁻¹, respectively) and, importantly, they both exhibited activity against H041 (MIC of 8 µg ml⁻¹) at a level comparable to that of trifluorinated phenol **9**, with **11** showing measurable binding to mosaic PBP2$^{H041}$ (Extended Data Table 3). The MICs for *R*,*R*-**12**, which is the active isomer of **10**, and both fluorinated single isomers, *R*,*R*-**13** and *R*,*R*-**14**, were 4 µg ml⁻¹ against H041 (Table 1 and Extended Data Table 3), indicating that an anion is preferred at the 4-position of the aryl glycine moiety.

Given that the introduction of anions improved activity, we next incorporated a phosphonate group onto the phenyl glycine moiety, beginning with a non-fluorinated *R*,*R*-**15** (Fig. 1). Strikingly, this resulted in a marked improvement in mosaic PBP2$^{H041}$ binding and antibacterial activity against both CRO-susceptible and -resistant strains in comparison to the phenol- and benzoate-containing boro-PBPi (Table 1). The 2,3-dioxopiperidine-containing analogues **15**–**17** demonstrated good binding to WT and mosaic PBP2 and excellent MICs of 0.06–0.12 and 0.5 µg ml⁻¹ against the ATCC 49226 and H041 strains, respectively (Extended Data Table 4).

Having discovered the importance of the terminal phosphonate group for activity, a series of boro-PBPi bearing the *N*-methylsulfonyl-2-imidazolinone motif present in mezlocillin was prepared (Supplementary Fig. 2). These boro-PBPi represented by

*R*,*R*-**18** had similar PBP2 binding affinity and antibacterial activity to the 2,3-dioxopiperidine-containing analogues (Fig. 1 and Table 1). Following the incorporation of multiple fluorine atoms (boro-PBPi **19**–**23**), *R*,*R*-**21** (VNRX-14079) was found to have two-to-eightfold greater potency than **18** against mosaic PBP2-producing strains (Table 1 and Extended Data Table 5). In addition, the MICs for **21** were eightfold lower than those for CRO against the CRO-resistant H041 and WHO Q strains that produce mosaic PBP2$^{H041}$ (encoded by *penA-37*) and PBP2$^{FC428}$ (encoded by *penA-60*), respectively (Table 1).

### Structural evidence for high-affinity binding of boro-PBPi to mosaic PBP2

Mutations in the transpeptidase (TPase) domain of PBP2 that are implicated in cephalosporin resistance in *N. gonorrhoeae*[28] seem to act by restricting the conformational dynamics of the protein[29,30]. We determined the crystal structures of the benzoate-containing **12** and phosphonate-containing **15** complexes in the mosaic PBP2 TPase domain derived from the extended-spectrum-cephalosporin-decreased susceptibility strain 35/02, tPBP2$^{35/02}$ (encoded by *penA-10*)[30], and of phosphonate-containing **21** in complex with tPBP2$^{H041}$ (Fig. 2). Essentially, complete electron density was observed for the inhibitors, except for weak density corresponding to the benzo ring of the benzoxaborinine system for the tPBP2$^{H041}$–**21** complex, and similarly for the boron-carbon bond in the tPBP2$^{35/02}$–**15** complex. In all structures, there is clear density indicating a covalent bond between the boro-PBPi boron atom and Ser310 of the tPBP2 active site. In addition, there are a number of hydrogen-bonding interactions common to all structures, including between the benzoxaborinine-associated carboxylate group and Thr498 and Thr500; Ser362 and the boronate ester oxygen atom; and Asn364 and the carbonyl group of the α-amido benzoxaborinine. Finally, the 2,3-dioxopiperidine rings of **12** and **15** and the *N*-methylsulfonyl-2-imidazolinone ring of **21** make similar stacking interactions with Tyr422.

Where interactions differ in structures is due to the altered substituent at the *para* position of the phenolic ring. In tPBP2$^{35/02}$, the *para*-carboxylate of **12** forms a hydrogen bond with Tyr543 in the $\beta_5$–$\alpha_{11}$ loop (Fig. 2a) and its replacement with a phosphonate residue in **15** (tPBP2$^{35/02}$ structure) and **21** (tPBP2$^{H041}$ structure) leads to the formation of additional hydrogen bonds with Arg502 and His514, which comprise the beginning and ending residues of the $\beta_3$–$\beta_4$ loop, in addition to that with Tyr543 (Fig. 2b,c).

A notable difference between **21** and both **12** and **15** is the addition of fluorines, where one is adjacent to the carboxylate of the benzoxaborinine system and another is on the phenyl ring adjacent to the phosphonate. In the crystal structure of tPBP2$^{H041}$ with **21**, the benzoxaborinine fluorine is largely exposed to solvent, with Tyr544 being the nearest residue, and unlikely to increase the affinity of the

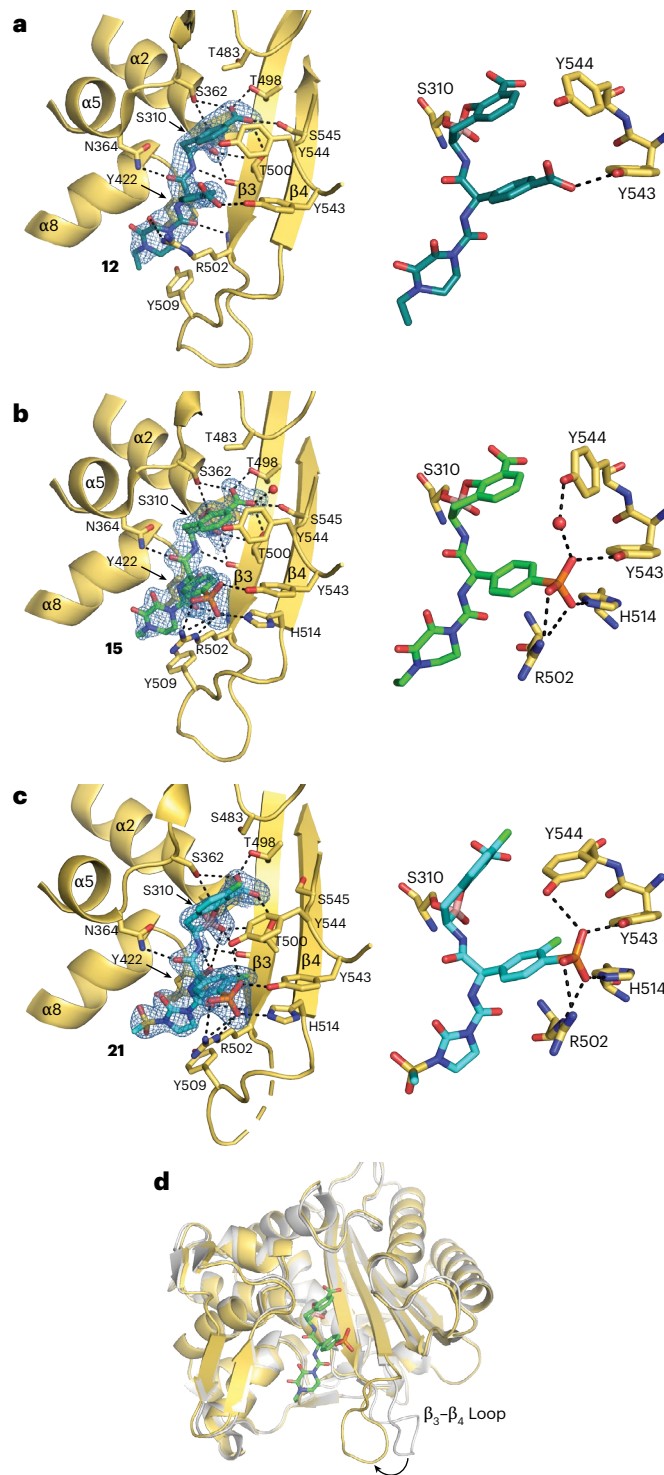

**Fig. 2 | Crystal structures of *N. gonorrhoeae* mosaic tPBP2 in complex with boro-PBPi 12, 15 and 21. a,b,** Crystal structures of tPBP2[35/02] bound to *R,R*-12 (a) and *R,R*-15 (b). **c,** Crystal structure of tPBP2[H041] bound to *R,R*-21. **a–c,** The right-hand panels show the interactions of PBP2 residues with the *para*-carboxylate of 12 (a), and the phosphonate of 15 (b) and 21 (c). **d,** Superimposition of the crystal structures of tPBP2[35/02] in complex with 15 and *apo* tPBP2[35/02] (Protein Data Bank: 6VBL) showing the movement of the β₃–β₄ loop towards the active site in the complex (arrow).

compound. In contrast, the fluorine on the phenolic ring projects within a hydrophobic core comprising the side chains of Thr500, Tyr543 and His514, where its presence may increase structural stability in this region.

The most striking conformational changes observed following boro-PBPi binding is the movement of the β₃–β₄ loop from a so-called 'outbent' conformation, as observed for *apo* structures of tPBP2[35/02] and PBP2[H041], to an inward conformation closer to the active site (Fig. 2d). The inward movement of the β₃–β₄ loop in the crystal structures in complex with the boro-PBPi is notable, as its failure to move in PBP2 variants derived from extended-spectrum-cephalosporin-resistant strains of *N. gonorrhoeae* is hypothesized to be an important contributor to CRO resistance[30]. The observed binding modes of 12, 15 and 21, with inward conformation of the β₃–β₄ loop, are similar to the complexes of tPBP2[H041] with cefoperazone and the ureidopenicillins piperacillin and azlocillin[31]. These β-lactams exhibit faster second-order acylation rates against tPBP2[H041] compared with the oxyimino-containing CRO[31]. These similar findings with the boro-PBPi are therefore consistent with the idea that higher-activity inhibitors work by overcoming the conformational barrier created by resistance mutations present in PBP2 from resistant strains.

To confirm the binding of 21 to the full-length PBP2, membranes were prepared from the cells of ATCC 49226 and H041, which carry non-mosaic and mosaic PBP2, respectively. As expected, 21 inhibited PBP2 and, interestingly, was specific for this PBP (Supplementary Fig. 3). At 1.2 µM, the binding to PBP2 from H041 was complete for 21 and partial for CRO (72% bound), indicating that 21 has higher affinity to the mosaic PBP2 than CRO, consistent with the binding affinity measured in the biochemical assay using tPBP2.

**Pharmacokinetic evaluation and in vivo efficacy of boro-PBPi**

Before performing murine efficacy studies, the plasma-protein-binding (PPB) and pharmacokinetic (PK) profiles of 18 and 21 were determined in mice. The PPB for 18 and 21 was 7 and 19%, respectively, in mouse plasma, whereas the PPB in human plasma was 24 and 39%, respectively. As shown in Supplementary Fig. 4, boro-PBPi 18 and 21 exhibited similar plasma concentration versus time profiles for each compound when administered to mice by either intravenous (IV) bolus or subcutaneous (SC) injection. Boro-PBPi 18 and 21 were well tolerated up to the highest dose tested (1,000 and 300 mg kg⁻¹ SC doses, respectively). Exposure, in terms of maximum concentration ($C_{max}$) and total overall exposure (area under the curve extrapolated from time 0 to infinity, $AUC_{inf}$), was less than dose proportional for 21 following SC administration at doses of 10–300 mg kg⁻¹, whereas for 18, exposure following SC injection was less than dose proportional for $C_{max} \geq 300$ mg kg⁻¹ and generally dose proportional for the $AUC_{inf}$ across all doses administered. The terminal half-life ($t_{1/2}$) of 18 was approximately 4 h following SC dosing with good exposure at all doses (Supplementary Table 1). The terminal $t_{1/2}$ of 21 was approximately 4 h at ≥100 mg kg⁻¹, whereas the terminal $t_{1/2}$ at 10 and 30 mg kg⁻¹ was not determined because concentrations at later time points were below the limit of quantification (Supplementary Table 2).

Assuming that all PBP-targeting drugs display time-dependent antibacterial activity in vivo as demonstrated for β-lactam-based PBP inhibitors[32,33], we selected the boro-PBPi dosages of 0.25, 7.50, 50 and 200 mg kg⁻¹ every 12 h (q12h) for the in vivo efficacy study in the mouse vaginal infection model with the CRO-susceptible FA1090 strain (agar MIC of 0.016 µg ml⁻¹ for 18 and 0.008 µg ml⁻¹ for CRO determined by CLSI guidelines[34]; Table 2). This model, available at Eurofins Inc., was used as proof-of-concept to examine in vivo efficacy of a boron-based PBP inhibitor against the drug-susceptible *N. gonorrhoeae* strain. In vivo efficacy of 18 was observed at ≥7.5 mg kg⁻¹ SC q12h, as PK simulations predicted that these doses would provide free unbound boro-PBPi concentrations in plasma above the MIC for at least 40% of the 24-h treatment duration (i.e. ≥40% PK-simulated *f*T > MIC); bacterial growth stasis was observed at 0.25 mg kg⁻¹ SC q12h (≤13% PK-simulated *f*T > MIC; Extended Data Fig. 1 and Supplementary Table 3). The efficacious doses of 18 were similar to those achieved by CRO at 1 mg kg⁻¹ (IV, single dose).

We next examined whether a boro-PBPi is efficacious in vivo against the more challenging CRO-resistant H041 strain than FA1090.

**Table 2 | Agar minimal inhibitory concentrations (µg ml$^{-1}$) of boro-PBPi and comparator agents against *N. gonorrhoeae* reference strains**

| Strain | PBP2 allele | 12 | 15 | 18 | 21 | CRO | ZOLI | GEPO | AZM | CIP | PEN |
|---|---|---|---|---|---|---|---|---|---|---|---|
| ATCC 49226 | *penA-22* | 0.5 | 0.06 | 0.06 | 0.03 | 0.016 | 0.12 | 0.25 | 0.25 | 0.004 | **0.5** |
| FA19 | *penA-15* | ≤0.016 | 0.008 | 0.008 | 0.008 | 0.004 | 0.03 | 0.12 | 0.06 | ≤0.002 | ≤0.016 |
| FA1090 | *penA-1* | 0.06 | 0.016 | 0.016 | 0.016 | 0.008 | 0.03 | 0.12 | 0.06 | 0.004 | 0.06 |
| WHO F | *penA-15* | ≤0.016 | ≤0.004 | 0.008 | 0.008 | ≤0.002 | 0.06 | 0.12 | 0.06 | 0.004 | ≤0.016 |
| WHO G | *penA-2* | 0.25 | 0.03 | 0.06 | 0.03 | 0.016 | 0.12 | 1 | 0.12 | 0.12[a] | 0.5[a] |
| WHO M | *penA-2* | 0.5 | 0.06 | 0.06 | 0.06 | 0.016 | 0.06 | 1 | 0.25 | 2[a] | >64[a] |
| MS11 | *penA-22* | 0.5 | 0.03 | 0.06 | 0.03 | 0.016 | 0.12 | 0.5 | 0.25 | 0.004 | 0.5[a] |
| WHO L | *penA-7* | 8 | 1 | 1 | 0.5 | 0.25 | 0.12 | 4 | 0.25 | 16[a] | 2[a] |
| WHO K | Mosaic *penA-10* | 4 | 0.25 | 0.5 | 0.12 | 0.12 | 0.12 | 0.25 | 0.25 | >16[a] | 2[a] |
| H041 | Mosaic *penA-37* | 16 | 1 | 1 | 0.25 | 1[a] | 0.12 | 0.5 | 0.25 | >16[a] | 2[a] |
| WHO Z | Mosaic *penA-64* | 2 | 0.25 | 0.25 | 0.06 | 0.5[a] | 0.12 | 0.25 | 1 | >16[a] | 2[a] |
| WHO Q (G7944) | Mosaic *penA-60* | 4 | 0.5 | 0.25 | 0.06 | 0.5[a] | 0.06 | 1 | >4[a] | 16[a] | 1[a] |
| CDC-0197 | Mosaic *penA-34* | 2 | 0.25 | 0.25 | 0.06 | 0.06 | 0.06 | 0.25 | >4[a] | 16[a] | 1[a] |
| F89 | Mosaic *penA-42* | >16 | >8 | >8 | >8 | 2[a] | 0.12 | 0.5 | 0.25 | 16[a] | 2[a] |

[a]Resistant/non-susceptible values based on the CLSI guideline[57], where MIC breakpoints are defined only for CRO, azithromycin, ciprofloxacin and penicillin G. AZM, azithromycin; CIP, ciprofloxacin; GEPO, gepotidacin; PEN G, penicillin G; ZOLI, zoliflodacin.

The in vivo efficacy of **21** was evaluated in a mouse vaginal infection model using the CRO-resistant H041 strain (agar MIC of 0.25 µg ml$^{-1}$ for **21** and 1 µg ml$^{-1}$ for both CRO and **18**; Table 2), performed at Uniformed Services University of the Health Sciences. Although these two infection models had different procedures in the pretreatment of mice and efficacy readout, the differences were not expected to affect the outcome of compound efficacy. Four dosing groups were selected to provide predicted %$f$T > MIC for **21** ranging from 20 to 100% when corrected for 19% mouse PPB. In this model, **21** demonstrated good efficacy against the CRO-resistant H041 strain, demonstrating a statistically significant reduction in the percentage of culture-positive mice relative to the vehicle control (*P* < 0.0001; Fig. 3 and Supplementary Table 4). Regimens of SC administration of 150 mg kg$^{-1}$ twice (q12h) or three times a day (q8h) resulted in 90 and 100% bacterial clearance by day 2, respectively, whereas a single SC dose (q24h) of 200 mg kg$^{-1}$ achieved 100% clearance by day 3. Results with 10 mg kg$^{-1}$ SC q12h were also significant, with 80% clearance by day 4 relative to the vehicle control (*P* < 0.01). The bacterial bioburden (colony forming units (CFU) ml$^{-1}$) in the vaginal swab suspension was consistent with bacterial clearance (percentage of mice cured of infection) in all treatment groups (Fig. 3). Based on the PK data and simulations, **21** administered as a single dose of 200 mg kg$^{-1}$ corresponded to a $f$T > MIC of 44% over 24 h, which was associated with clearance of H041 and consistent with the efficacy of **18** against FA1090 (≥40% $f$T > MIC). Notably, neither the evolution of spontaneous gonococcal mutants nor a significant change in vaginal polymorphonuclear leukocytes was observed between groups during this in vivo study (Supplementary Fig. 6). No adverse effects were observed in mice treated with any dose of **21**.

**Microbiological activity of boro-PBPi 21**

Compound **21** showed comparable MICs to CRO against strains producing non-mosaic PBP2 and lower MICs against mosaic PBP2-producing strains (Table 2). The latter included the CRO-non-susceptible strain WHO Q, which belongs to the globally expanding FC428 lineage[35,36]. The WHO M strain that produces TEM-1 β-lactamase was susceptible to **21**, providing support for the stability of benzoxaborinines against β-lactamase inactivation. The strains WHO L (containing non-mosaic PBP2) and F89 (also known as WHO Y, containing mosaic PBP2) were the least susceptible strains to **21**. WHO L carries a PBP2 A501V variant (Supplementary Table 7) that reduces CRO susceptibility[37,38] and

probably affects the activity of **21**. F89 has mosaic PBP2 (*penA*-42), which is identical to PBP2 (*penA*-34) of the CDC-0197 strain but with an additional A501P alteration (Supplementary Table 7). The difference in MIC against these strains indicates that the A501P mutation impacts the activity of multiple boro-PBPi, including **21**. Notably, strains carrying the PBP2 A501P variant have not spread since their emergence in 2010 (refs. 39,40), probably owing to the fitness cost of producing this mosaic PBP2 (ref. 41). Although A501P mutation in PBP2 is rare, PBP2 variants such as PBP2$^{WHOL}$ (carrying the A501V substitution) are found in circulating lineages resistant to extended-spectrum cephalosporins[42–44]. These data indicate that specific variants of PBP2 reduce the activity of **21**. Additionally, through the use of isogenic strains, we confirmed that the PorB porin and the Mtr efflux system minimally affect the activity of boro-PBPi (Supplementary Table 8).

Zoliflodacin and gepotidacin are topoisomerase/gyrase inhibitors that have been recently approved for the treatment of gonorrhoea[45–47]. Zoliflodacin, which had activity comparable to that of **21** against the mosaic PBP2-producing strains (except F89, as stated in the previous paragraph), was the most active comparator agent; gepotidacin was slightly less active than zoliflodacin against the strain panel (Table 2). Because the molecular target of boro-PBPi is PBP2, no cross-resistance to these topoisomerase/gyrase inhibitors is expected. To confirm this, we isolated mutants with reduced susceptibility to zoliflodacin from WHO K (all mutants possessing GyrB D429N) and found no cross-resistance to **21** in these mutants (Supplementary Table 9).

Boro-PBPi were tested against the CDC panel of 44 *N. gonorrhoeae* clinical isolates circulating in the US. Compound **21** had MIC$_{50}$ and MIC$_{90}$ values of 0.06 and 0.12 µg ml$^{-1}$, respectively, which were equivalent to CRO and slightly more active than zoliflodacin against these strains (Extended Data Table 6). As expected from the primary strain panel data, **18** had fourfold higher MIC$_{50}$ and MIC$_{90}$ values than **21**, whereas the MIC$_{50}$ and MIC$_{90}$ of benzoate-containing **12** were eightfold higher than that of **18**. In time-kill assays using three strains (ATCC 49226, WHO Q and H041), both **21** and CRO reduced the bacterial bioburden by ≥2log$_{10}$ in 6 h and below the limit of detection at 24 h relative to the untreated controls (Supplementary Fig. 7), confirming similar levels of bactericidal activity of **21** compared with CRO. The frequencies of resistance to **21** and CRO, determined on agar containing 4× and 16× MIC of each agent, were <5 × 10$^{-9}$ against ATCC 49226, WHO Q, H041, WHO L and CDC-0197 (Supplementary Table 11).

To examine whether mutants with reduced susceptibility to **21** could arise at sub-MIC levels of **21**, we used wild-type ATCC 49226, two strains carrying PBP2 with a substitution at the Ala501 residue (WHO L and CDC-0197) and the CRO-resistant globally spreading strain (WHO Q). Susceptibility to **21** or CRO did not change over ten passage cycles in ATCC 49226. The susceptibility of WHO Q to CRO decreased gradually, yet its growth attenuated at day 2 and ceased at day 4 when the cells were passaged with **21**. Although their susceptibility to CRO did not change, the susceptibility of WHO L and CDC-0197 to **21** decreased at day 7 and the mutants isolated on day 8 had elevated MIC of **21** (Extended Data Fig. 2). Genome sequence analyses revealed that the mutant isolated from CDC-0197 contained a mutation causing PBP2 Y543C substitution (Supplementary Table 12). Given that Tyr543 interacts with the phosphonate of **21**, this substitution may reduce the binding affinity in this mosaic PBP2. Notably, a BLAST search indicated that no clinical strain carrying PBP2 with the Y543C substitution has been isolated to date. In the mutant isolated from WHO L, no alteration in PBP2 was identified, while other mutations causing deletion of genes including *mtrE* that encodes the MtrCDE efflux pump were detected.

## Boro-PBPi 21 exhibits promising safety and selectivity properties

Compound **21** demonstrated no haemolytic activity at 1 mg ml$^{-1}$, no mammalian cytotoxicity at 256 µg ml$^{-1}$, no mitochondrial toxicity (IC$_{50}$ > 100 µM) and no chromosomal aberrations (IC$_{50}$ > 300 µM) in micronucleus assays at 100 µM (Extended Data Table 7 and Supplementary Table 13). When tested at 30 µM, no noteworthy inhibition of seven cytochrome P450 enzymes and low-level inhibition (34.6% decrease in baseline activity) of CYP2C19 were observed for **21**. Because boro-PBPi inhibits PBP2 by forming a covalent bond with the active site serine nucleophile, the activity of **21** was assessed at 30 µM against major mammalian serine proteases (chymotrypsin, trypsin and thrombin A); negligible inhibition was noted, suggesting selectivity for inhibition of PBP2 (Extended Data Table 7 and Supplementary Table 13). In addition, no noteworthy binding to the hERG channel was observed for **21** at 30 µM. These results suggest favourable antibacterial selectivity, low potential for drug–drug interactions and minimal cardiac channel-mediated toxicity. Furthermore, **21** had a good aqueous

solubility (>700 µM) at physiological pH and was stable in both plasma and hepatocytes (Extended Data Table 7 and Supplementary Table 13), implicating renal excretion as the main route of elimination for **21**. Further work, such as an animal PK studies using isotopically labelled drug, will need to be conducted for precise determination of the route of elimination.

Intramuscular administration is recommended for an injectable small-molecule therapeutic agent to treat gonorrhoea in the outpatient setting[48]. In the aforementioned mouse in vivo efficacy studies, we used SC administration because IM administration in mice is challenging due to the small injection volume compared with the volume for SC administration[49]. We therefore evaluated the potential for IM administration of **21** in male Sprague Dawley rats (Extended Data Fig. 3). Based on the PK results, the IM bioavailability of **21** was estimated to be at least 66%, providing support for the potential of IM administration of **21** for the treatment of gonorrhoea.

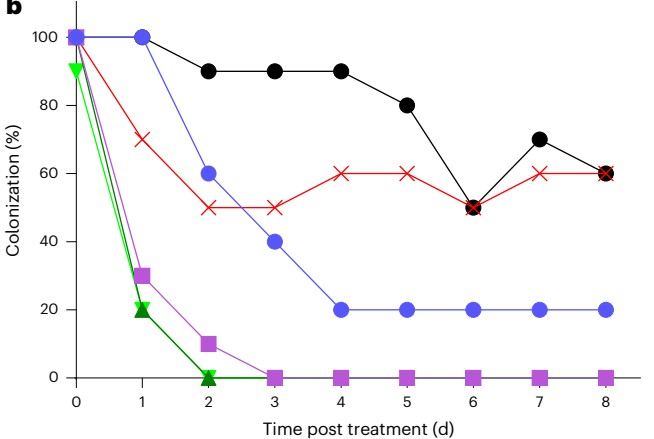

**a**

| | Group | Test article | Dosing on day 1 | Total dose (mg kg$^{-1}$) | Predicted % $f$T/MIC* |
|---|---|---|---|---|---|
| ●(blue) | | **21** | 10 mg kg$^{-1}$, SC, q12h | 20 | 20 |
| ■(purple) | | **21** | 200 mg kg$^{-1}$, SC, q24h | 200 | 44 |
| ▲(dark green) | | **21** | 150 mg kg$^{-1}$, SC, q12h | 300 | 76 |
| ▼(green) | | **21** | 150 mg kg$^{-1}$, SC, q8h | 450 | 100 |
| ✕(red) | | CRO | 120 mg kg$^{-1}$, SC, q8h | 360 | |
| ●(black) | | Vehicle (PBS) | 5 ml kg$^{-1}$, SC, q8h | | |

*Boro-PBPi 21 MIC = 0.25 µg ml$^{-1}$ against H041

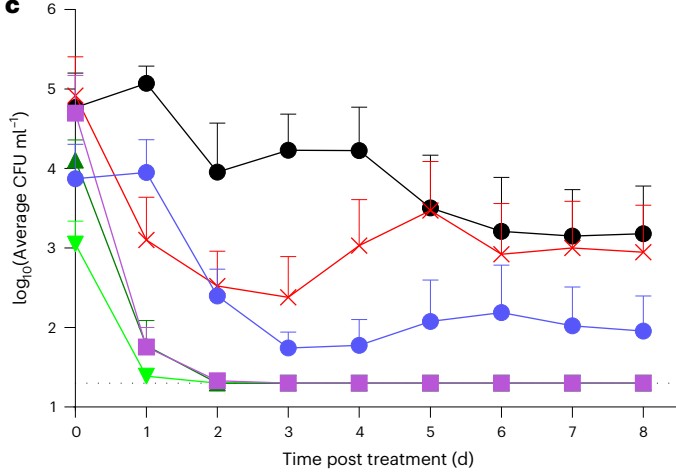

**Fig. 3 | In vivo efficacy of boro-PBPi 21 in the mouse vaginal infection model of ceftriaxone-resistant *N. gonorrhoeae* H041. a**, Dosing regimen to examine the in vivo efficacy of **21** against the CRO-resistant H041-STM$^R$ strain in the mouse vaginal infection model with the predicted % $f$T/MIC. Compound **21** was administered only on day 1. Vaginal swab samples from infected mice ($n = 10$) were cultured to confirm H041 infection on day 0 before the administration of the test compound or controls on day 0 (green box). Vaginal cultures were collected on days 1–8 post treatment to determine the bacterial burden (CFU ml$^{-1}$). **b**, Percentage of mice infected over time with H041 and the average bacterial burden recovered over the course of the experiment. More than 85% of the vehicle control mice remained colonized through day 5, with 60% colonized by day 8. Subcutaneously administered CRO, used as a positive control, yielded unanticipated underperformance compared with traditional intraperitoneal injection[33]. **c**, The average bacterial burden recovered over the course of the experiment. Error bars indicate the s.e.m. ($n = 10$ mice). The horizontal dashed line denotes the limit of detection (20 CFU ml$^{-1}$). $P$ values for comparison of groups were calculated using a two-way analysis of variance of the log$_{10}$-transformed values ($F$ (5, 54) = 13.34). When monitoring the growth of commensals on heart infusion agar plates, bacterial growth was observed for 0–20% of the mice in the untreated and CRO-treated control groups as well as the test group that received boro-PBPi **21** at 10 mg kg$^{-1}$ q12h, 200 mg kg$^{-1}$ q24h and 150 mg kg$^{-1}$ q8h. The majority of isolates were Gram-positive cocci in clusters and some isolates were Gram-positive diplococci, probably *Staphylococcus* sp. and *Enterococcus* sp., respectively. In the test group administered 150 mg kg$^{-1}$ q12h, 60% of the mice were colonized with *Staphylococcus*-like bacteria. Staphylococcal presence in vaginal microbiota usually does not affect the duration of gonococcal colonization or the bioburden. **b,c**, Groups colour-coded as in **a**. Source bacterial burden data and $P$ values are provided.

## Discussion

During the discovery phase of this programme we identified compound **2**, a prototype boro-PBPi containing the ureido side chain of cefoperazone, with weaker antibacterial activity than cefoperazone against *N. gonorrhoeae*. This reduced activity may be a consequence of the reversible nature of the covalent bond between **2** and the catalytic Ser310 of PBP2, whereas cefoperazone and other β-lactams form an irreversible acyl-enzyme complex with PBP2. We subsequently identified benzoate-containing compound **12** and aryl phosphonate-containing compounds **15**, **18** and **21** with increased interactions with PBP2 and antibacterial activity. The preference for phosphonate-containing boro-PBPi is partly related to the ability of the phosphonate to interact with PBP2 residues Arg502, His514, Tyr543 and Tyr544, thus stabilizing the boro-PBPi–PBP2 complex. Binding of **12**, **15** and **21** elicits an inward conformation of the β$_3$–β$_4$ loop, which is associated with higher acylation of β-lactams to PBP2 (refs. 31,50). The importance of the β$_3$–β$_4$ loop movement for potent boro-PBPi activity is also supported by the observation that Ala501 mutations, known to cause ordering of the β$_3$–β$_4$ loop[38], reduced boro-PBPi activity in WHO L (A501V mutation) and F89 (A501P mutation). Importantly, the antibacterial activities of these ureido boro-PBPi were less impacted by altered PBP2 variants (for example, mosaic PBP2) than oxyimino-acetamide boro-PBPi (for example, compound **1**), which is consistent with the observation that ureidopenicillins and ureidocephalosporins are more potent inhibitors of mosaic PBP2 than the oxyimino-acetamide cephalosporins[31].

Although the acquisition of an Ala501 mutation in PBP2 (for example, A501P in F89) may lead to boro-PBPi resistance, the A501P mutation is associated with reduced biological fitness in vitro and in vivo[41], and strains containing this mutation have not been reported in over a decade. In contrast, *penA-60* has been observed globally in many CRO-resistant clinical isolates. This variant has similar mutations to PBP2 from H041 except for a Thr at position 316 in place of Pro in H041. Only minor fitness defects are observed for *penA-60*-possessing strains[51], which have probably contributed to their rapid spread. The MIC of **21** against WHO Q, which carries the *penA-60* allele, was 0.06 μg ml$^{-1}$, eightfold lower than the CRO MIC. Similar to CRO, **21** showed a low frequency of resistance (<$4.17 \times 10^{-9}$) and rapid cell killing at 4× MIC in WHO Q, consistent with no resistant mutant arising with ten passages at sub-MIC concentrations. Thus, **21** is active against CRO-resistant *N. gonorrhoeae* strains carrying the *penA-60* allele and retains favourable drug-like properties akin to CRO.

Overall, the lead boro-PBPi **21** exhibited potent stand-alone antibacterial activity against *N. gonorrhoeae*, good safety and selectivity profiles, and excellent in vivo efficacy. It is therefore poised to be a promising agent for the treatment of gonorrhoea infections including those resistant to CRO. The next steps for advancing **21** are process development for scale-up synthesis, extended microbiological testing (including evaluation of susceptibility to large strain collections and effects on cell aggregation[52]), the characterization of PK/efficacy exposure-response relationships, comprehensive in vitro safety and absorption, distribution, metabolism and excretion (ADME) assessments, rodent/non-rodent toxicology studies and assessment of projected costs. If all are successful, **21** will follow a traditional preclinical and clinical development course. Together, compound **21** is an outstanding candidate to replace CRO for the outpatient treatment of gonorrhoea.

## Methods

### Compounds and chemical synthesis

Boro-PBPi compounds were synthesized at Venatorx Pharmaceuticals, Inc. and BioDuro-Sundia as described in the 'Chemistry experimental details' section in the Supplementary Information. Ceftriaxone (USP, 1098184), azithromycin (USP, 1046056), ciprofloxacin (Sigma-Aldrich, 17850), cefoperazone (Alfa Aesar, J65185), zoliflodacin (MedChemExpress, HY-17647) and gepotidacin (MedChemExpress, HY-16742) were used. The Molecular Operating Environment (Chemical Computing Group) software was used for computational chemistry, inhibitor design and the calculation of physicochemical properties, including p$K_a$.

### Bacterial strains and growth conditions

The strains used in this study are listed in Supplementary Table 14. FA1090, H041 (WHO X), MS11 and F89 were obtained from A.E.J. (Uniformed Services University of the Health Sciences). The *N. gonorrhoeae* WHO reference strains (WHO F, WHO G, WHO L, WHO M, WHO K, WHO L, WHO Y and WHO Z)[53,35] were obtained from the CDC. WHO Q (G7944)[36] was obtained from the National Collection of Type Cultures (NCTC 14208). FA19 was obtained from R. A. Nicholas (University of North Carolina). A set of 50 CDC strains (AR bank, 165–214)[54] was obtained from the CDC, of which six strains were confirmed not to be *N. gonorrhoeae* and were excluded from the study. All *N. gonorrhoeae* strains were cultured at 37 °C in a 5% CO$_2$ humidified environment.

### Whole-genome sequencing and analysis

Whole-genome sequencing and genomic analyses were conducted on the parent and isolated mutant strains. DNA extraction, Illumina library preparation and Illumina sequencing (paired read with 2 × 150 bp) were performed by GENEWIZ from Azenta Life Sciences. Whole-genome-sequencing analysis of the FASTQ files provided by GENEWIZ was performed using Geneious Prime versions 2022.1.1 and 2025.1.3 (Biomatters Inc.). The reads were trimmed using BBDuK Adapter/Quality Trimming version 38.84 (Brian Bushnell) to yield approximately ten million reads. The trimmed reads were mapped to the corresponding reference genomes (GenBank accession numbers provided in Supplementary Table 15) and genetic alterations were identified, especially in the *penA* gene encoding PBP2 (the major target of CRO), genes encoding the major porin (PorB) and the efflux pump gene cluster (*mtrR*–*mtrCDE*). For CDC strains[54] whose assembled genomes were not available, sequence reads were assembled to generate contigs that were used as the reference genome. To identify genetic alterations in the genome sequences of mutants compared with those of the parent strain, trimmed reads were mapped to the corresponding reference genome sequences (Supplementary Table 15) and differences in genetic alterations between the parent and mutant strains were identified. A BLAST search was performed against the NCBI database (https://blast.ncbi.nlm.nih.gov/) using short amino acid sequences containing the identified PBP2 Y543C substitution to determine the presence of any PBP2 proteins with the Y543C substitution. The raw reads are deposited in GenBank under BioProject accession number PRJNA1353147.

### Minimum inhibitory concentration assays

To determine the ability of test compounds to inhibit the growth of bacterial strains, MIC assays were performed using both liquid- and agar-based microdilution methods, as described previously[34,55,56], with modifications. Briefly, cryo-preserved bacterial cultures of clinical strains were streaked for isolation and cultured for 24 h on Chocolate Agar. The Chocolate Agar was made by first separately autoclaving 72 g l$^{-1}$ (2×) GC agar base (Remel, R453502) and 2% (2×) haemoglobin (Remel, R451402) at 121 °C for 20 min for sterilization. After cooling to approximately 50 °C, the 2×GC agar base and 2×haemoglobin solutions were combined in a 1:1 vol/vol ratio and 1% IsoVitaleX Enrichment (BD, 211876) was added to the solution. Freshly cultured colonies were used for inoculum preparation. In the liquid broth-based MIC assay, twofold serial dilutions of the test compounds were made in a 96-well plate with a final volume of 75 μl per well at twofold the final desired concentration in Fastidious Broth (FB; prepared according to Cartwright et al. with a final pH 7.2 ± 0.2)[55]. To make the assay inoculum, a direct suspension was prepared by aseptically swabbing 10–15 colonies from the agar plates into culture tubes containing 2 ml fresh sterile saline. After the dilution plates were set up, direct suspensions were diluted in a cuvette containing sterile saline and the optical density at 600 nm (OD$_{600}$) was measured. Direct suspensions were diluted in FB to make assay inocula

(approximately $1 \times 10^6$ CFU ml$^{-1}$) and 75 μl assay inocula were added to the 96-well plates prepared with compound dilutions, yielding a starting bacterial concentration of $5 \times 10^5$ CFU ml$^{-1}$. The plates were incubated for approximately 24 h. The broth MIC was determined visually as the lowest concentration that completely inhibited bacterial growth.

For the agar-based method, GC agar was used to determine the MIC. After the initial strain incubation for about 24 h, bacterial colonies were re-streaked for isolation on fresh Chocolate Agar and cultured for 18–24 h. To prepare GC agar, the GC agar base (Remel, R453502) dissolved in water was autoclaved at 121 °C for sterilization and once cooled to approximately 50 °C, it was supplemented with IsoVitaleX Enrichment (BD, 211876) and 1 M filter-sterilized ferric nitrate (Fisher Scientific, S25320) to achieve final concentrations of 1% and 12 μM, respectively. The determined amount of test compound stock solution was pipetted directly into a 100 mm × 15 mm Petri dish for each desired test concentration and 20 ml of the prepared GC agar (before solidification) was added to the Petri dish containing the test compound. The agar plates were swirled to dissolve the compound and vented for solidification in a biological safety cabinet until dry. For the preparation of assay inoculum, a direct suspension was first prepared by aseptically swabbing several colonies from the agar plates and suspending them in 2 ml sterile cation-adjusted Mueller–Hinton broth (CAMHB); these were then diluted with sterile CAMHB and adjusted to an OD$_{600}$ of 0.1 (0.5 McFarland standard equivalent). A Steer's replicator was used to plate up to 32 spots of 2 μl inocula on a single agar plate from a separate 32-well plate containing inocula (500 μl each). The agar plates were vented in the biological safety cabinet until they were dry, inverted and incubated for 24 h. The agar MIC was determined as the lowest concentration that completely inhibited the bacterial growth.

### Time-kill assay
The time-kill assay for *N. gonorrhoeae* was performed as described[57], with the following modifications. The bacterial inoculum for the assays was prepared by suspending colonies of *N. gonorrhoeae* cultured on Chocolate Agar for approximately 24 h in sterile saline. The cell suspension was adjusted to an OD$_{600}$ of 0.1 and diluted 1:500 in FB broth (final cell density of approximately $1 \times 10^5$ CFU ml$^{-1}$; 90 μl each) in a 96-well plate, which was then incubated with shaking at 200 rpm for 4 h to allow the bacteria to reach the logarithmic phase of growth. After the initial growth phase, 10 μl of the appropriate 10× drug dilution was added to each well (total volume of 100 μl) to achieve final drug concentrations of 4× and 16× MIC. For the untreated control, 10 μl FB was added. Samples were taken at −4 h (time of inoculation), 0 h (time of compound addition) as well as 2, 4, 6 and 24 h following the addition of the compound and the CFU ml$^{-1}$ was determined for each sample. Growth curves were analysed by plotting the $\log_{10}$(CFU ml$^{-1}$) versus time using GraphPad Prism version 10.3.2.

### Frequency of resistance
*N. gonorrhoeae* strains (ATCC 49226, H041 and WHO Q) were cultured on Chocolate Agar for about 24 h and colonies were suspended (approximately $5 \times 10^8$ CFU ml$^{-1}$) in PBS. A 0.1-ml aliquot of the cell suspension was spread on ten GC agar plates (100 mm diameter) supplemented with 1% IsoVitaleX, 12 μM ferric nitrate and test compounds (4× and 16× MIC) for ATCC 49226 and H041. Chocolate Agar was used for WHO Q because the strain grew poorly on GC agar. The viable cell count in each suspension was determined by plating serial tenfold dilutions onto the corresponding agar. The plates were incubated for 24 h, the visible colonies were counted and spontaneous mutation frequency was calculated.

### Isolation of zoliflodacin-resistant mutants and mutation identification
Zoliflodacin-resistant mutants were isolated by selection of WHO K (MIC = 0.12 μg ml$^{-1}$) on GC agar supplemented with 4× MIC of zoliflodacin. Colonies arising on agar were streaked on Chocolate Agar and

a single colony was tested for broth MIC to confirm the elevated MIC of zoliflodacin. Short-read genome sequencing was performed on the parent and isolated mutant strains (GENEWIZ). Genome analysis from FASTQ files provided by GENEWIZ was performed using Geneious Prime version 2022.1.1 (Biomatters Inc.). The reads were trimmed using BBDuK Adapter/Quality Trimming version 38.84, yielding approximately ten million reads. Read mapping was performed with the Geneious Mapper using the WHO K reference genome (GenBank ID: GCF_900087865.2) to provide >100 mean coverage of the entire chromosome, followed by Geneious single-nucleotide-polymorphism analysis, resulting in the identification of the GyrB(D429N) mutation present in each mutant.

### Serial passage experiments with ceftriaxone and boro-PBPi 21
Glycerol stocks of the strains tested (ATCC 49226, WHO L, CDC-0197 and WHO Q) were cultured on fresh Chocolate Agar for approximately 24 h at 37 °C with 5% CO$_2$. The arising bacterial colonies were re-streaked on fresh Chocolate Agar for second isolation and cultured for 18–24 h for use in the assay. To prepare the assay inoculum, a direct suspension was made by aseptically swabbing several colonies from the agar plates and suspending them in 2 ml sterile CAMHB; the direct suspensions were then diluted with sterile CAMHB and adjusted to an OD$_{600}$ of 0.1 (0.5 McFarland standard equivalent). Each inoculum (10 μl) was spotted on GC agar (0.6 ml; 3–4 mm depth) containing the desired concentrations (0.25, 0.50, 1, 2, 4 and 8× agar MIC) of CRO and **21** in 24-well flat-bottomed plates. After the plates were incubated for 20–24 h at 37 °C with 5% CO$_2$, susceptibility was recorded as the lowest concentration that completely inhibited bacterial growth. For passaging with the compound, cells that grew robustly on agar containing the highest concentration of each compound were swabbed, suspended in 0.5–1 ml fresh CAMHB and the suspension was adjusted to an OD$_{600}$ of 0.1. The cell suspension (10 μl) was spotted on GC agar containing the compound in 24-well plates, followed by incubation for 20–24 h at 37 °C with 5% CO$_2$. Passaging experiments were serially performed ten times (one passage per day) and susceptibility was monitored daily. After days 1 and 10 as well as any day when a change in susceptibility was observed, the agar MIC was determined as described earlier.

### Expression and purification of *N. gonorrhoeae* PBP2
In the *N. gonorrhoeae* PBP2 inhibition assay, constructs comprising the TPase domain of PBP2 from the FA19 and H041 strains (tPBP2$^{FA19}$ and tPBP2$^{H041}$) were used. The cloning, expression and purification of these PBP2 constructs were previously described[29,58]. Importantly, the TPase domains of PBP2 were acylated by fluorescently labelled penicillin V, Bocillin-FL (Thermo Fisher Scientific) at the same rate as that of full-length PBP2 (ref. 58).

### Measurement of in vitro inhibition of *N. gonorrhoeae* PBP2
The inhibitory potency towards wild-type PBP2$^{FA19}$ was assessed by determining the inhibitor concentration required to reduce PBP2$^{FA19}$ binding to Bocillin-FL by 50% (IC$_{50}$) using a fluorescence polarization competitive equilibrium binding assay[59]. Enzyme titration/saturation binding experiments were initially performed to establish the assay conditions for competitive binding. A solution of 0.2 μM Bocillin-FL in assay buffer (50 mM HEPES–NaOH pH 8.0), 300 mM NaCl and 10% (vol/vol) glycerol) was prepared and saturation binding was performed by mixing 40 μl PBP2 at different concentrations (0–24 μM) with 40 μl of the 0.2 μM Bocillin-FL solution in individual wells of black 384-well microplates. The fluorescence was monitored immediately following mixing using a Cytation 3 plate reader (BioTek) with a fluorescence polarization cube containing polarizing filters with 485 nm excitation and 520 nm emission. The instrument gain was set to achieve minimum fluorescence values of 50–60 and each measurement was the average of three flashes. Fluorescence was measured continuously for up to 120 min and the response was stabilized within 10 min, with a

dose-dependence on the PBP2$^{FA19}$ concentration. For the competition binding assays, twofold serial dilutions of compounds were prepared in assay buffer and mixed with Bocillin-FL (final concentration of 0.1 μM) in black 384-well microplates. PBP2$^{FA19}$ was added to a final concentration of 0.25 μM and fluorescence was immediately measured at 1-min intervals for up to 10 min. Fluorescence values at the reaction end point (typically within 8 min) were normalized to the maximal response plotted against inhibitor concentration and fitted to a four-parameter inhibitor-response curve to derive the IC$_{50}$.

As the interaction between PBP2$^{H041}$ and Bocillin-FL did not produce an adequate response in the fluorescence polarization assay, the inhibitory potency towards tPBP2$^{H041}$ was assessed by determining the concentration of the compound required to inhibit the binding of Bocillin-FL to PBP2$^{H041}$ by 50% (IC$_{50}$) using an SDS–PAGE-based competition binding assay. Twofold serial dilutions of each compound were prepared in the assay buffer, mixed with 1 μM tPBP2$^{H041}$ and incubated at ambient temperature for 60 min. Bocillin-FL (1 μM) was added and the reaction mixtures were further incubated for 60 min before resolution by SDS–PAGE using 10% NuPAGE Bis-Tris mini protein gels (Invitrogen). The amount of Bocillin-FL incorporated into tPBP2$^{H041}$ at each inhibitor concentration was detected by fluorescence scanning of the SDS–PAGE gels with an Azure 600 imager (Azure Biosystems, Inc.) and quantified using the ImageJ software[60]. Data were normalized to the maximum fluorescence, plotted against the inhibitor concentration and fitted to a four-parameter inhibitor-response curve to derive the IC$_{50}$.

## Membrane preparation and PBP-binding assay using the membrane

Membrane preparation and PBP-binding assays were performed as previously described[61,62], with modifications. Glycerol stocks of the ATCC 49226 and H041 strains were cultured on fresh Chocolate Agar for approximately 24 h at 37 °C with 5% CO$_2$. The arising bacterial colonies were picked and suspended in FB to an OD$_{600}$ of approximately 0.7, which was inoculated with 1% into 325 ml FB supplemented with 1/100 volume 4.2% sodium bicarbonate. The culture was incubated at 37 °C with shaking at 160 rpm until an OD$_{600}$ of 0.6–0.8 was reached and cooled on ice. The cells were harvested by centrifugation at 5,000$g$ and 4 °C for 10 min, washed twice with ice-cold PBS and maintained frozen at −80 °C. The cells were resuspended in PBS, sonicated and centrifuged at 15,000$g$ and 4 °C for 30 min. The supernatant was ultracentrifuged at 136,000$g$ and 4 °C for 1 h, washed and the pellet was resuspended in 0.5 ml PBS. The protein concentration of the membrane was approximately 1 mg ml$^{-1}$.

In the PBP-binding assay, the solutions containing 8 μg of the membrane and 1 μl of threefold serially diluted inhibitors (CRO and boro-PBPi 21 at final concentrations of 0, 0.4, 1.2, 3.6, 11, 33, 100 and 300 μM) were incubated at room temperature for 30 min, followed by the addition of 1 μl of 100 μM Bocillin-FL (final concentration of 10 μM) and incubation at room temperature for 30 min. Finally, the samples were mixed with 3.3 μl of 4×SDS sample buffer and incubated at 45 °C for 30 min. The PBPs were separated on SDS–PAGE gels and visualized by fluorescence (excitation, 472 nm; emission, 513 nm) using an Azure 600 imager (Azure Biosystems). The IC$_{50}$ value of each inhibitor was determined using Fiji ImageJ[63].

## X-ray crystallography

The crystal structure of the transpeptidase domains of PBP2 derived from the CRO-reduced-susceptibility strain 35/02 of *N. gonorrhoeae* (tPBP2$^{35/02}$) and resistant strain H041 has been reported[30]. Proteins were purified and concentrated to 13 mg ml$^{-1}$ in Tris–HCl (pH 7.8) with 10% glycerol and 500 mM NaCl, and then crystallized in the same way, as reported[30]. These crystals occupy the P2$_1$2$_1$2$_1$ space group with one molecule in the asymmetric unit. Complexes of tPBP2$^{35/02}$ or tPBP2$^{H041}$ with 12, 15 and 21 were generated by soaking crystals with 1 μl of 10 mM boro-PBPi dissolved in PBS for 4 h (12 and 15) or 5–10 min (21),

followed by flash-freezing in liquid nitrogen. For 15, a mixture of the *S,R*- and *R,R*-diastereomers was used for soaking; only the *R,R* form was observed in the electron density. For complexes with tPBP2$^{35/02}$, diffraction data were collected at a wavelength of 1.00 Å on an Eiger-16M detector at the SER-CAT 22-ID beamline at the Advanced Photon Source (Argonne, IL, USA); 200° of data were collected in 0.25° oscillations with an exposure time of 0.2 s per frame and a crystal-to-detector distance of 200 mm. For the tPBP2$^{H041}$–21 complex, data were collected at a wavelength of 0.978 Å on an Eiger2 9M detector at the NYX 19-ID beamline (National Synchrotron Light Source II). A total of 180° of data were collected in 0.25° oscillations with 0.1 s exposure per frame and a crystal-to-detector distance of 200 mm. The datasets were processed using HKL2000 (ref. 64) and structures were solved by refinement against the *apo* structure of tPBP2$^{35/02}$ or tPBP2H$^{H041}$. Bound inhibitors were modelled into the |F$_o$| − |F$_c$| difference electron density map, followed by iterative cycles of model building using COOT[65] and refinement with PHENIX[66] for the tPBP2$^{35/02}$ structures and REFMAC5 (ref. 67) for tPBP2$^{H041}$. The crystallographic data collection and model refinement statistics are listed in Supplementary Table 16.

## Approval of facilities and procedures for animal studies

The facilities and procedures for animal studies were accredited by the Association for Assessment and Accreditation of Laboratory Animal Care International and assured from the Office of Laboratory Animal Welfare. All animal rooms used for the PK and efficacy studies were maintained at a temperature range of 20–24 °C and humidity of 30–70%, with 12 h light–dark cycles, unless otherwise specified.

## Pharmacokinetic analysis of boro-PBPi 18 in mice

Boro-PBPi 18 was formulated at a maximum concentration of 100 mg ml$^{-1}$ in PBS adjusted to pH 6–7 using NaOH. The PK study was conducted at the BioDuro-Sundia DMPK group (Jiangsu, China) using female BALB/c mice (7–9 weeks old; Vital River Laboratories). The study protocol (BioDuro BD-202102114) was approved by the Institutional Animal Care and Use Committee (IACUC). Boro-PBPi 18 was administered once via IV injection (3 mg kg$^{-1}$) or SC injection at 10, 30, 100, 300 or 1,000 mg kg$^{-1}$ (three mice per group, allocated randomly). Blood microsamples (30 μl each) were collected at following the time points following administration: 0.083, 0.25, 0.5, 1, 2, 4, 8 and 24 h. Plasma from each K$_2$EDTA-treated whole-blood sample was prepared using centrifugation at 2,000$g$ and 4 °C for 5 min. The concentration of boro-PBPi in the plasma was measured using an ultra-performance liquid chromatography–tandem mass spectrometry (UPLC–MS/MS) platform (Waters Acquity UPLC and Sciex 6500). Each plasma sample was diluted in 5% trichloroacetic acid in water, followed by centrifugation at 2,000$g$ and room temperature for 15 min. The supernatant was diluted twofold with 50% acetonitrile and loaded onto an Avantor ACE 5 C4 column (50 mm × 2.1 mm). Mobile phase (MP) A was 5 mM ammonium acetate and 0.05% formic acid in water and MP B was 0.1% formic acid in acetonitrile. Compound 18 was eluted in a linear gradient from 5% to 95% MP B over 1.2 min at a flow rate of 0.6 ml min$^{-1}$ and the elution was subjected to MS/MS analysis using multiple reaction monitoring in positive-ion mode with a transition of 611.10 to 429.00. Buspirone (5 ng ml$^{-1}$) and tolbutamide (50 ng ml$^{-1}$) in 5% trichloroacetic acid were used as the internal standard. The lower limit of quantification for 18 was 2 ng ml$^{-1}$.

## Pharmacokinetic analysis of boro-PBPi 21 in mice

Boro-PBPi 21 was formulated at a maximum concentration of 30 mg ml$^{-1}$ in PBS adjusted to pH 6–7 using NaOH. The PK study was conducted at the BioDuro-Sundia DMPK group using male CD-1 mice (7–9 weeks old; Vital River Laboratories). The IACUC-approved protocol number was BioDuro BDW-2201-0007. The PK study of 21 was conducted in a manner similar to that of 18. The bioanalytical method used to quantify 21 was performed using a UPLC–MS/MS platform (ExionLC

AD system and a SCIEX 7500). Each sample prepared from the plasma was loaded onto an ACQUITY UPLC peptide BEH C18 column (1.7 µm, 300 Å, 50 mm × 2.1 mm). MP A was 1% formic acid in water and MP B was 1% formic acid in acetonitrile. Compound **21** was eluted at a flow rate of 0.5 ml min$^{-1}$ in a linear gradient from 5% to 55% of MP B over 0.1 min and an isocratic hold at 55% MP B for 0.8 min, followed by a linear gradient to 95% of MP B over 0.1 min; the elution was subjected to MS/MS analysis using multiple reaction monitoring in positive-ion mode with a transition of 647.4 to 465.0. Tolbutamide (0.5 ng ml$^{-1}$ in 5% trichloroacetic acid) was used as an internal standard. The lower limit of quantification for **21** was 2 ng ml$^{-1}$.

The parameters derived from non-compartmental PK analysis were as follows: plasma concentration at time 0 after IV administration ($C_0$), plasma $t_{1/2}$, plasma $C_{max}$, AUC of the concentration versus time curve from $t = 0$ to infinity ($AUC_{inf}$), volume of distribution at steady-state ($V_{SS}$), clearance (CL), time to maximum plasma concentration ($T_{max}$) and bioavailability from the SC dose ($F_{SC}$). In each of these instances, variability was related to concentrations below the lower limit of quantification, which restricted the characterization of the terminal phase in at least one mouse per group (Supplementary Table 2).

### Efficacy assessment of boro-PBPi 18 in the FA1090 vaginal infection model

Efficacy analysis was performed by Eurofins Pharmacology Discovery Services, through support by CARB-X and the Pre-Clinical Services group at the National Institute of Allergy and Infectious Diseases, using ovariectomized and 17β oestradiol-treated female BALB/c mice[68]. The study protocol (PDS IM005-07302018) was approved by the IACUC. Groups of five immunocompetent female ovariectomized BALB/c mice (5–6 weeks old) were used for the efficacy study. The mice were obtained from BioLASCO Taiwan and acclimated for three days. After acclimation, an ovariectomy was performed on 4-week-old mice. The period of surgical recovery and acclimation was at least seven days. The mice were treated with meloxicam (SC injection; 20 mg kg$^{-1}$) if signs of pain or distress were observed during this period. Before infection, the mice were administered SC injections of oestradiol solution (0.23 mg per mouse) two days before infection (day −2) as well as on the day of infection (day 0). To minimize the indigenous vaginal bacteria, the animals were treated (q12h) with streptomycin (1.2 mg per mouse) and vancomycin (0.6 mg per mouse) by intraperitoneal injection along with 0.4 mg ml$^{-1}$ trimethoprim sulfate supplied in the drinking water. These antibiotic treatments started two days before infection and continued daily until the end of the study. On day 0, the mice were inoculated intravaginally with *N. gonorrhoeae* FA1090 (ATCC 700825; 0.02 ml (2.56 × 10$^6$ CFU) per mouse) under anaesthesia induced by intraperitoneal injection of 80 mg kg$^{-1}$ pentobarbital, followed by rinsing the vagina with 50 mM HEPES (pH 7.4; 30 µl). The mice were administered SC injections of **18** twice a day with a 12 h interval (q12h) starting at 2 h post infection for one day (five mice per group, allocated randomly). At 2 or 26 h after infection, the mice were killed by CO$_2$ asphyxiation to harvest their vaginal lavage fluid. For vaginally infected mice, vaginal lavage was performed twice with 200 µl GC broth containing 0.05% saponin to recover vaginal bacteria. Lavage samples from each mouse were pooled in a total volume of 500 µl. Bacterial burden in the lavage fluids was determined by performing tenfold serial dilutions and plating 0.1 ml of each dilution onto Chocolate Agar plates. The bacterial burden (CFU ml$^{-1}$) of the lavage fluid was calculated.

### Efficacy assessment of boro-PBPi 21 in the H041 vaginal infection model

The in vivo efficacy of **21** was tested in female NCI BALB/c mice (6–7 weeks old) in A.E.J.'s laboratory at the Uniformed Services University, as described[33]. The mice were obtained from Charles River Laboratories. This study protocol was approved by the Uniformed Services University IACUC under protocol MIC23-759. Using a simulation based on the mouse PPB (19%) and the PK profile in male CD-1 mice, four dosing regimens for **21** were selected to vary %$f$T > MIC from 20% to 100% in this efficacy study. Mice in the dioestrus or anoestrus stage of the oestrous cycle were randomized into six groups and pretreated with 17β oestradiol as well as antibiotics (streptomycin and trimethoprim) to suppress the overgrowth of the commensal microbiota and to support *N. gonorrhoeae* colonization. The pretreated mice were vaginally infected with H041 (1–2 × 10$^4$ CFU per mouse) two days before treatment with **21** via SC injection, either once or alternate dosing regimens. Control groups were administered CRO (120 mg kg$^{-1}$ SC, q8h) or vehicle control (PBS). Vaginal mucus was quantitatively cultured for *N. gonorrhoeae* for eight consecutive days post treatment. A portion of the swab was also inoculated onto heart infusion agar to monitor the presence of facultative aerobic commensal microbiota. Vaginal polymorphonuclear leukocyte influx was assessed on each culture day by cytological examination of stained vaginal smears and reported as the percentage of polymorphonuclear leukocytes among 100 vaginal cells. Efficacy was measured by comparing differences in the clearance rate and average bacterial burden over eight days following treatment with **21**, CRO or compound vehicle PBS.

### In vitro safety and selectivity assays for boro-PBPi 21

The assays that examined PPB, cytotoxicity, haemolysis, mitochondrial toxicity and chromosomal aberrations (micronucleus assay) as well as several in vitro inhibition assays (CYP450s and human proteases) or binding (hERG) assays for **21** were performed using established methods and controls that behaved as expected. Eight cytochrome P450 enzymes (CYP1A2, CYP2C19, CYP2D6, CYP3A4, CYP2E1, CYP2B6, CYP2C19 and CYP2C8), recommended by the US FDA[69], were used to predict metabolism-mediated drug–drug interactions. A detailed description of the methods and raw data are provided in Supplementary Table 13.

### Pharmacokinetic analysis of boro-PBPi 21 in Sprague Dawley rats

The PK study was performed in male Sprague Dawley rats (5–7 weeks old; Hilltop Labs Animals, Inc.) after a single IV or IM dose at QPS, LLC. This study protocol was approved by the IACUC under QPS IACUC protocol number 005. The designated animal room where the rat PK study was performed was maintained with 12 h light–dark cycles at a temperature range of 20–26 °C and relative humidity of 20–70%. All dosing formulations of **21** were freshly prepared as clear solutions. For IM dose preparation, **21** was dissolved in vehicle (45:55 (vol/vol) 1 N NaOH:1.6×PBS) by vortexing, sonication and stirring to achieve a concentration of 75 mg ml$^{-1}$ at pH 6.8. A 0.6 mg ml$^{-1}$ IV dose formulation was prepared by diluting the IM formulation with 1×PBS (pH 7.4; approximately final pH 7.1) and filtering through a 0.22-µm filter (Millex-GV). Male Sprague Dawley rats (weight of approximately 270 g) were administered **21** via IM and IV routes. Blood samples (0.3–0.4 ml) were collected via the tail vein at pre-dose (0 h) and 0.083, 0.25, 0.5, 2, 4, 8 and 24 h post treatment from IV-dosed rats ($n = 3$), and pre-dose (0 h) and 0.5, 1, 2, 4, 8 and 24 h post treatment from IM-dosed rats ($n = 3$ rats per group, allocated randomly). All blood samples were collected into tubes containing K$_2$EDTA on wet ice and centrifuged at approximately 3,800$g$ and 4 °C for 3 min within 40 min of blood collection. Terminal blood samples were collected in 10-ml Vacutainer tubes and centrifuged at 3,300$g$ and 4 °C for 15 min. All plasma samples were snap-frozen on dry ice and stored at approximately −70 °C until bioanalysis to determine the concentration of **21** in plasma. The PK data (Extended Data Fig. 3) were evaluated using non-compartmental analysis (Phoenix WinNonlin, version 8.3, Certara USA Inc.) to determine the IM bioavailability in rats. The average concentrations from three rats per dose group were included. The terminal $t_{1/2}$, $R^2$ and adjusted $R^2$ values were 0.63 h, 0.98 and 0.98, respectively, following IV administration of 3 mg kg$^{-1}$ **21** to rats, and 0.95 h, 0.93 and 0.89, respectively, following IM administration of 30 mg kg$^{-1}$ **21**. The IM bioavailability

was calculated by dividing the dose-normalized $AUC_{inf}$ based on the last observed concentration (observed $AUC_{inf}$ dose in h kg ng ml$^{-1}$ mg$^{-1}$) when administered IM by the observed $AUC_{inf}$ when administered IV (i.e. 66% (1030.81 / 1558.40)).

## Statistics and reproducibility
No statistical method was used to pre-determine sample size. No data were excluded from the statistical analyses.

## Reporting summary
Further information on research design is available in the Nature Portfolio Reporting Summary linked to this article.

## Data availability
The data that support the findings of this study are included in Article and its Supplementary Information. Structural data have been deposited in the Protein Data Bank under the accession codes 9MD0 (tPBP2$^{35/02}$–12), 9MCZ (tPBP2$^{35/02}$–15) and 9Z5T (tPBP2$^{H041}$–21). Raw sequence reads obtained from *N. gonorrhoeae* strains in this study were deposited in GenBank under BioProject accession number PRJNA1353147. Source data are provided with this paper.

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

## Acknowledgements

We thank our former colleagues at Venatorx Pharmaceuticals, Inc., for helpful discussions, and R. A. Nicholas (University of North Carolina), W. Shafer (Emory University), E. Brown (McMaster University) and J.-D. Docquier (University of Siena) for providing strains. BioDuro-Sundia, Eurofins Panlabs Discovery Services Taiwan, Ltd and QPS are the contract research organizations that supported this work, including compound synthesis, mouse PK, DMPK, in vivo efficacy and rat PK. We also thank M. Barbachyn, A. Erwin and S. Chiang (CARB-X) for supportive discussions as well as our former colleagues at Venatorx for preliminary data generation and helpful discussions. The research reported in this publication was supported by the National Institute of Allergy and Infectious Diseases of the National Institutes of Health under award number R01 AI141239 (D.M.D.), and by CARB-X under research subaward agreement number 4500003206 (S.M.C.). The X-ray crystallography work was supported in part by the National Institutes of Health award R01 AI164794 (C.D.). The content is the sole responsibility of the authors and does not necessarily represent the official views of the National Institutes of Health and the Uniformed Services University, the Henry M. Jackson Foundation for the Advancement of Military Medicine or the Department of Defense. CARB-X's funding for this project is provided in part with federal funds from the US Department of Health and Human Services (HHS), Administration for Strategic Preparedness and Response, Biomedical Advanced Research and Development Authority under agreement number 75A50122C00028 as well as awards from Wellcome (WT224842) and Germany's Federal Ministry of Education and Research (BMBF). The content of this publication is solely the responsibility of the authors and does not necessarily represent the official views of CARB-X or any of its funders. X-ray crystallography data were collected at the Southeast Regional Collaborative Access Team (SER-CAT) 22-ID beamline at the Advanced Photon Source, Argonne National Laboratory. SER-CAT is supported by its member institutions (https://www3.ser.aps.anl.gov/contact-us#TITLE_SER_CAT_Memberships), equipment grants (S10_RR25528, S10_RR028976 and S10_OD027000) from the National Institutes of Health, and funding from the Georgia Research Alliance. This research used the resources of the Advanced Photon Source, US Department of Energy (DOE) Office of Science user facility operated for the DOE Office of Science by Argonne National Laboratory under contract number DE-AC02-06CH11357. The NYX beamline 19-ID at the National Synchrotron Light Source II is supported by the New York Structural Biology Center. This research also used resources of the National Synchrotron Light Source II, a US DOE Office of Science User Facility operated for the DOE Office of Science by Brookhaven National Laboratory under contract number DE-SC0012704. The NYX Eiger2 detector was supported by grant S10OD030394 through the Office of the Director, National Institutes of Health.

## Author contributions

T.U., A.L.Z., B.M., L.M.A., S.A.B., C.L.C., G.-H.C., A.S.D., M.E., S.G.E.H., N.J.L., C.L.M., G.R., A.S., K.U., F.Y., D.C.P., C.J.B., D.M.D. and S.M.C. contributed to compound synthesis and data generation on antibacterial activity, in vitro PBP binding, ADME, toxicity, PK evaluations, in vivo efficacy study and/or the rat PK study. B.W., Z.L., M.W., Z.Z., X.Z. and H.Z. contributed to the compound synthesis. T.U., S.A.B., C.M.S., S.B. and C.D. contributed to structural data generation and analyses. R.T. and A.E.J. contributed to the in vivo efficacy study using the H041 infection model. T.U., A.L.Z., B.M., S.A.B., C.L.M., D.M.D. and S.M.C. wrote the manuscript with contributions from other authors.

## Competing interests

T.U., A.L.Z., B.M., L.M.A., S.A.B., C.L.C., G.-H.C., A.S.D., M.E., S.G.E.H., N.J.L., C.L.M., G.R., A.S., K.U., F.Y., D.C.P., C.J.B., D.M.D. and S.M.C. are former employees of Venatorx Pharmaceuticals, Inc. T.U., A.L.Z., S.A.B., C.L.C., G.-H.C., N.J.L., C.L.M., D.C.P., C.J.B., D.M.D. and S.M.C. are co-inventors of a patent application covering the molecules described in this manuscript (US20240270762, EP4347607, CN117693511, AU2022279901, CA3219907). D.C.P. is the President and CEO of Spring Mill Pharma, Inc., which currently owns the intellectual property of the boro-PBPi programmes run at Venatorx Pharmaceuticals, Inc. B.W., Z.L., M.W., Z.Z., X.Z. and H.Z. are current or former employees of BioDuro-Sundia. C.M.S., S.B., C.D., R.T. and A.E.J. declare no conflicts of interest.

## Additional information

**Extended data** is available for this paper at https://doi.org/10.1038/s41564-026-02309-3.

**Correspondence and requests for materials** should be addressed to Tsuyoshi Uehara or Stephen M. Condon.

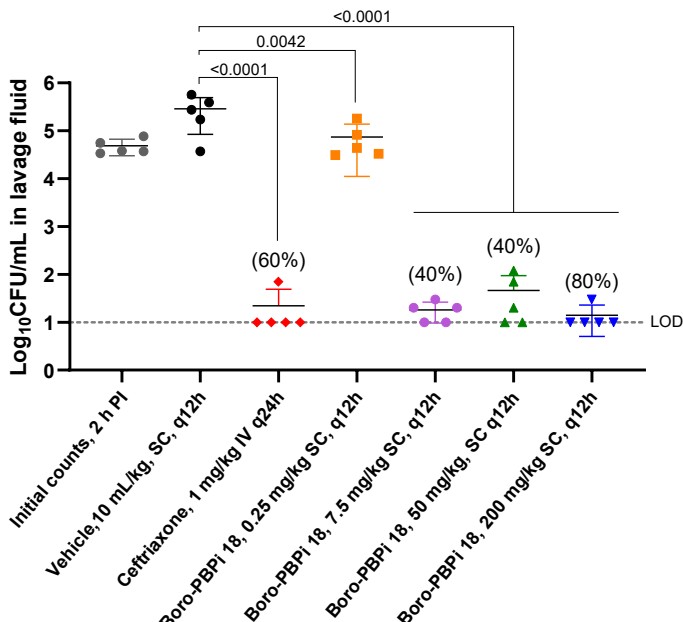

**Extended Data Fig. 1 | In vivo efficacy result for boro-PBPi 18 in the murine vaginal infection model with ceftriaxone-susceptible *N. gonorrhoeae* FA1090 (ATCC 700825).** Significance was calculated relative to the CFU ml$^{-1}$ with the vehicle at 26 h and assessed using one-way ANOVA (F (6, 28) = 8.503, $n$ = 5 mice).

The *P* values for the comparison of antibiotic-treated samples with the vehicle control are shown. The percentage (%) shown in the figure represents the percentage of animals with bacterial counts below the limit of detection (LOD, dotted line). Source numerical data are provided.

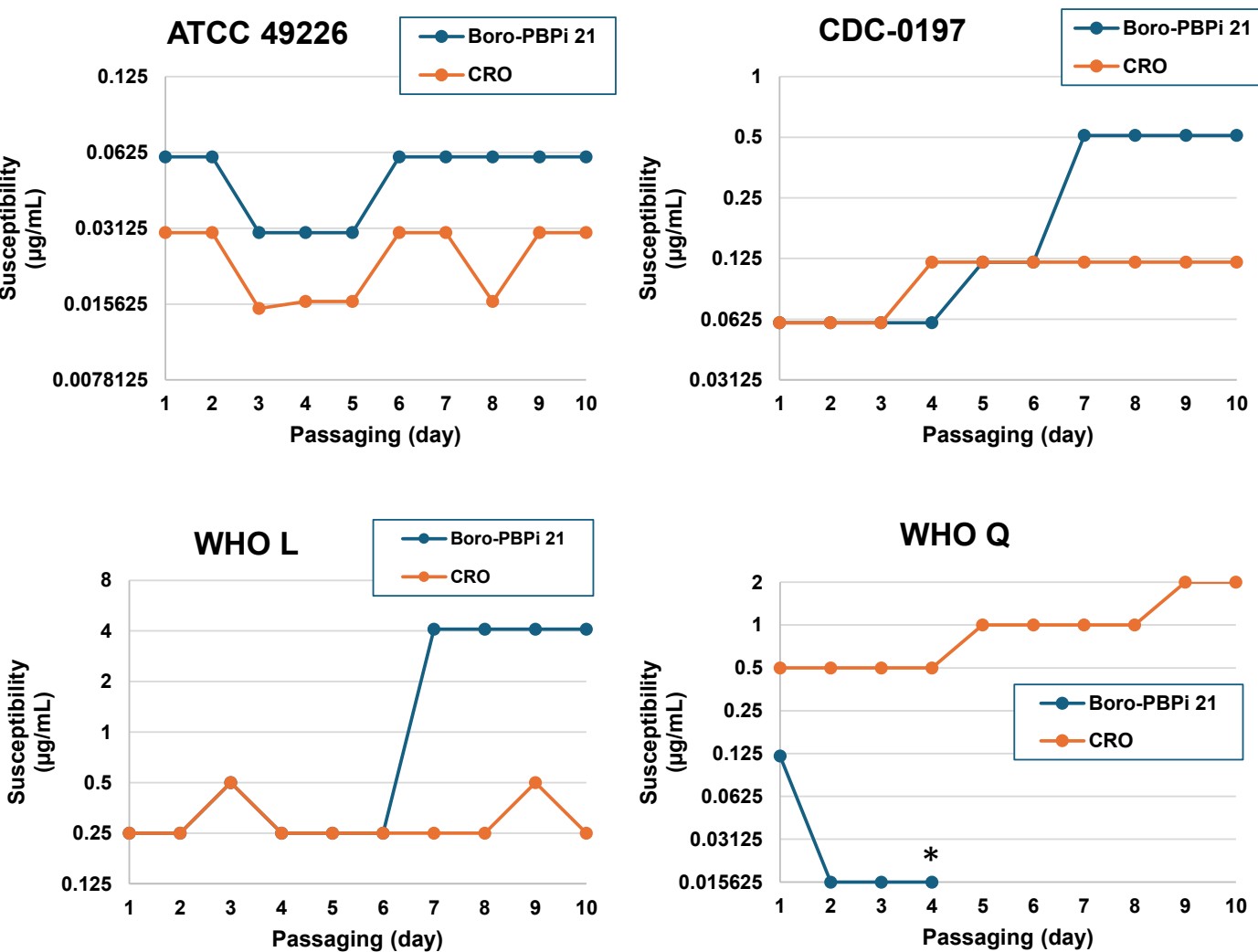

**Extended Data Fig. 2 | Ten-day passaging of ATCC 49226, CDC-0197, WHO L and WHO Q with sub-MIC of boro-PBPi 21 and CRO.** The plots on the top show the susceptibility of cells passaged with **21** and CRO each day. The passaging of WHO Q with **21** terminated at day 4 due to no growth observed in all wells (asterisk).

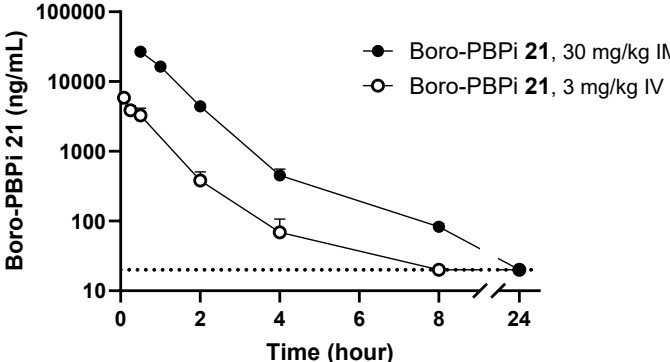

**Extended Data Fig. 3 | Plasma pharmacokinetics of boro-PBPi 21 in rats.**
Data are presented as mean values ± s.d. The dotted line shows the detectable quantitation level (concentration above 20 ng ml⁻¹). The bioanalytical conditions are described in Supplementary Tables 17–19. The concentration of **21** in the plasma PK study was quantified by a LC-MS/MS system with a Waters ACQUITY UPLC I-Class PLUS System and a Waters TQS-micro tandem mass spectrometer. Each sample prepared from the plasma was loaded onto a Waters ACQUITY Premier BEH C18 column (1.7 μm, 300 Å, 50 mm × 2.1 mm). MP A was 0.5% formic acid in water and MP B was 0.5% formic acid in acetonitrile. **21** was eluted at a 0.5 ml min⁻¹ flow rate in linear gradients from 15% to 30% of MP B over 0.7 min and to 95% of MP B over the next 0.7 min, and the elution was subjected to the MS/MS analysis using MRM in positive-ion mode with a transition of 647.04 → 629.10. Levofloxacin was used as an internal standard. Source numerical data are provided.

**Extended Data Table 1 | Broth microdilution MIC (µg ml⁻¹) of ceftriaxone and boro-PBPi against multiple bacterial species**

| Compound | *Ec* ATCC 25922 (QC strain) | *Ec* BAS901C* (*lptD4123*) | *Ec* D22* (*lpxC101*) | *Kp* UMM (KPC-2) | *Pa* ATCC 27853 (QC strain) | *Pa* ATCC 35151* | *Ab* ATCC 19606 (Type strain) | *Ng* ATCC 49226 (non-mosaic PBP2) | *Ng* H041 (mosaic PBP2) | *Sa* ATCC 29213 (MSSA) |
|---|---|---|---|---|---|---|---|---|---|---|
| **Ceftriaxone** | 0.06 | ≤0.015 | 0.03 | >128 | 16 | 0.06 | 32 | 0.008 | 1 | 4 |
| **1** | 32 | 8 | 16 | 32 | >128 | 32 | >128 | 16 | >128 | 8 |
| **2** | 32 | 1 | 8 | 32 | >128 | 2 | >128 | 4 | 64 | >128 |
| **12** | 32 | 0.06 | 4 | 32 | 32 | 0.25 | 128 | 0.5 | 4 | >128 |
| **15** | 32 | 0.06 | 4 | 32 | 64 | 0.5 | 16 | 0.06 | 0.5 | >128 |
| **18** | 64 | 0.06 | 4 | 64 | >128 | 2 | 16 | 0.12 | 1 | >128 |
| **21** | >128 | 0.5 | 16 | >128 | >128 | 4 | 128 | 0.06 | 0.12 | >128 |

*Hyperpermeable strain. The description of the bacterial strains used are listed in Supplementary Table 14. Abbreviations: Ec, *E. coli*; Kp, *K. pneumoniae*; Pa, *Pseudomonas aeruginosa*; Ab, *Acinetobacter baumannii*; Ng, *N. gonorrhoeae*; Sa, *Staphylococcus aureus*.

**Extended Data Table 2 | Wild-type (WT) PBP2 binding and minimal inhibitory concentration (MIC) data for selected phenol-containing benzoxaborinines**

| Compound |  | | | | Biological activity | | |
|---|---|---|---|---|---|---|---|
| | | | | | IC$_{50}$, µM | MIC, µg/mL | |
| | R$^1$ | R$^3$ | R$^4$ | R$^5$ | WT *Ng* PBP2$^{WT}$ | ATCC 49226 | H041 (WHO X) |
| cefoperazone | NA | NA | NA | NA | 0.3 | 0.06 | 0.25 |
| 2 | H | H | H | H | 2.8 | 4 | 64 |
| *R,R-3 | F | H | H | H | ND | 2 | 32 |
| *R,R-4 | F | H | H | F | ND | 16 | 16 |
| 5 | H | H | F | H | 1.7 | 8 | 32 |
| 6 | H | F | F | H | 0.7 | 2 | 32 |
| 7 | F | F | H | H | 0.3 | 8 | 32 |
| 8 | F | H | F | H | 1.8 | 8 | 32 |
| 9 | F | F | F | H | 0.8 | 1 | 8 |

*Single diastereomer. ND, not determined. PBP2$^{WT}$, PBP2 from the FA19 strain.

**Extended Data Table 3 | PBP2 binding and MIC data for selected benzoate-containing benzoxaborinines**

| Compound | R¹ | R³ | R⁴ | R⁵ | IC₅₀(*Ng* PBP2), μM | | MIC, μg/mL | |
|---|---|---|---|---|---|---|---|---|
| | $R^1$ | $R^3$ | $R^4$ | $R^5$ | PBP2$^{WT}$ | PBP2$^{H041}$ | ATCC 49226 | H041 (WHO X) |
| **10** | H | H | H | H | 1.2 | ND | 1 | 8 |
| **11** | H | H | H | F | ND | 395 | 4 | 8 |
| *R,R*-**12** | H | H | H | H | 1.0 | 42 | 0.5 | 4 |
| *R,R*-**13** | F | H | H | H | 1.2 | ND | 1 | 4 |
| *R,R*-**14** | H | F | F | H | 0.8 | 33 | 0.5 | 4 |

*Single diastereomer. ND, not determined.

**Extended Data Table 4 | PBP2 binding and MIC data for selected 2,3-dioxopiperidine-containing benzoxaborinines**

| Compound | R$^1$ | R$^3$ | R$^4$ | R$^5$ | IC$_{50}$(*Ng* PBP2), µM | | MIC, µg/mL | |
|---|---|---|---|---|---|---|---|---|
| | | | | | PBP2$^{WT}$ | PBP2$^{H041}$ | ATCC 49226 | H041 (WHO X) |
| *R,R*-**15** | H | H | H | H | 0.7 | 7.1 | 0.06 | 0.5 |
| *R,R*-**16** | F | H | H | H | 0.3 | 2.5 | 0.06 | 0.5 |
| *R,R*-**17** | H | H | H | F | 1.4 | 10.0 | 0.12 | 0.5 |

**Extended Data Table 5 | PBP2 binding and MIC data for selected _N_-methylsulfonyl-2-imidazolinone-containing benzoxaborinines**

| Compound |  | | | | Biological activity | | | |
|---|---|---|---|---|---|---|---|---|
| | | | | | IC$_{50}$(_Ng_ PBP2), µM | | MIC, µg/mL | |
| | R$^1$ | R$^3$ | R$^4$ | R$^5$ | PBP2$^{WT}$ | PBP2$^{H041}$ | ATCC 49226 | H041 (WHO X) |
| _R,R_-**18** | H | H | H | H | 0.3 | 3.2 | 0.12 | 1 |
| _R,R_-**19** | F | H | H | H | 0.6 | 4.7 | 0.06 | 0.5 |
| _R,R_-**20** | F | F | H | H | 1.3 | 2.1 | 0.12 | 0.5 |
| _R,R_-**21** | F | H | H | F | 1.7 | 3.0 | 0.06 | 0.12 |
| _R,R_-**22** | H | F | F | H | 0.5 | 3.7 | 0.12 | 1 |
| _R,R_-**23** | H | F | F | F | 2.6 | ND | 0.06 | 0.25 |

ND, not determined.

**Extended Data Table 6 | MIC50/MIC90 and MIC distribution of boro-PBPi and competitor compounds against 44 CDC isolates**

| Compound | MIC$_{50}$ | MIC$_{90}$ | Number of strains at agar MIC (µg/ml) | | | | | | | | | |
|---|---|---|---|---|---|---|---|---|---|---|---|---|
| | | | ≤0.016 | 0.03 | 0.06 | 0.12 | 0.25 | 0.5 | 1 | 2 | 4 | ≥8 |
| Ceftriaxone | 0.06 | 0.12 | 4 | 2 | 25 | 12 | 1 | | | | | |
| 12 | 2 | 4 | | | 2 | 1 | 1 | | 2 | 22 | 15 | 1 |
| 15 | 0.25 | 0.5 | 3 | 1 | | 3 | 21 | 16 | | | | |
| 18 | 0.25 | 0.5 | 3 | 1 | | 3 | 16 | 21 | | | | |
| 21 | 0.06 | 0.12 | 3 | 1 | 21 | 18 | 1 | | | | | |
| Zoliflodacin | 0.12 | 0.12 | | | 8 | 32 | 4 | | | | | |
| Gepotidacin | 0.5 | 1 | | | | | 7 | 21 | 16 | | | |
| Azithromycin | 0.5 | 1 | | | | 1 | 18 | 17 | 4 | | 1 | 3 |
| Ciprofloxacin | 16 | 16 | 5 | | | | | | 1 | 7 | 2 | 29 |
| Penicillin G | 2 | 2 | | | | 3 | | | 12 | 25 | 2 | |

MIC values for each strain are listed in Supplementary Table 10.

**Extended Data Table 7 | Safety, selectivity and stability of boro-PBPi 21**

| In vitro ADMET assay | Boro-PBPi 21 results |
|---|---|
| Solubility | 75 mg ml⁻¹ in PBS adjusted to pH 6.5 – 7 (formulated), >700 μM in PBS pH 7.4 (thermodynamic solubility) |
| Plasma protein binding (% bound) | 39% (human), 19% (mouse) |
| Haemolysis | >1,000 μg ml⁻¹ (human red blood cells) |
| Cytotoxicity | $CC_{50}$ >256 μg ml⁻¹ (HeLa, MRC-5, 3T3) |
| Mitochondrial toxicity | No toxicity observed at 100 μM in a Glu/Gal assay using SKOV-3 human ovarian cancer cells |
| CYP inhibition | $IC_{50}$ >30 μM for 8 CYP enzymes* |
| Micronucleus assay (mammalian chromosomal aberration assay) | No micronucleus observed up to 300 μM in CHO-K1 cells with and without liver S9 fraction pre-incubation |
| hERG cardiac channel assay | No inhibition at 30 μM |
| Protease inhibition | Negligible inhibition of three human proteases at 30 μM (18% inhibition for chymotrypsin observed) |
| Plasma stability | Half-life: >6 h (human), >6 h (mouse) |
| Hepatocyte metabolism stability | Half-life: >6 h (human), >6 h (mouse) |

The in vitro ADMET assays are described in Supplementary Table 13. *With testing eight cytochrome P450 enzymes to predict metabolism-mediated drug–drug interactions, no noteworthy inhibition (≥10% decrease in baseline activity) of seven cytochrome P450 enzymes and low-level inhibition (34.6% decrease in baseline activity) of CYP2C19 were observed.

# Reporting Summary

## Statistics

For all statistical analyses, confirm that the following items are present in the figure legend, table legend, main text, or Methods section.

| n/a | Confirmed | |
|---|---|---|
| ☐ | ☒ | The exact sample size (*n*) for each experimental group/condition, given as a discrete number and unit of measurement |
| ☐ | ☒ | A statement on whether measurements were taken from distinct samples or whether the same sample was measured repeatedly |
| ☐ | ☒ | The statistical test(s) used AND whether they are one- or two-sided *Only common tests should be described solely by name; describe more complex techniques in the Methods section.* |
| ☐ | ☒ | A description of all covariates tested |
| ☐ | ☒ | A description of any assumptions or corrections, such as tests of normality and adjustment for multiple comparisons |
| ☐ | ☒ | A full description of the statistical parameters including central tendency (e.g. means) or other basic estimates (e.g. regression coefficient) AND variation (e.g. standard deviation) or associated estimates of uncertainty (e.g. confidence intervals) |
| ☐ | ☒ | For null hypothesis testing, the test statistic (e.g. *F*, *t*, *r*) with confidence intervals, effect sizes, degrees of freedom and *P* value noted *Give P values as exact values whenever suitable.* |
| ☒ | ☐ | For Bayesian analysis, information on the choice of priors and Markov chain Monte Carlo settings |
| ☒ | ☐ | For hierarchical and complex designs, identification of the appropriate level for tests and full reporting of outcomes |
| ☒ | ☐ | Estimates of effect sizes (e.g. Cohen's *d*, Pearson's *r*), indicating how they were calculated |

*Our web collection on statistics for biologists contains articles on many of the points above.*

## Software and code

Policy information about availability of computer code

**Data collection**   X-ray diffraction data were collected at the SER-CAT 22-ID beamline at the Advanced Photon Source in Argonne, IL, USA.

**Data analysis**
HKL2000; X-ray diffraction data processing
PHENIX 1.18.2-3874 Industrial Consortium Map sharpening; model refinement and validation
COOT 0.9.8.95 MRC Laboratory of Molecular Biology, Computational Structural Biology Group; Model building
REFMAC5 5.7.0009 MRC Laboratory of Molecular Biology, Computational Structural Biology Group; Refinement of Macromolecular Structures
ChemDraw Professional 22 and 23; chemical analysis
Molecular Operating Environment (MOE) 2022 and 2024; Structure/ligand-based inhibitor design, SAR, modeling, computational chemistry
Microsoft 365 (Excel, PowerPoint); Data analyses, tables, figures
GraphPad Prism 9 and 10; Data plotting and analyses, nonlinear regression, Anova analysis
Phoenix WinNonlin 8.3; Pharmacokinetics analysis
Geneious Prime 2022, 2023, 2024, 2025, and 2026; DNA and protein sequence analysis including genome sequence analysis
Waters Analyst 1.7.1 and SCIEX OS 2.0.0.45330; UPLC-MS/MS data acquisition

For manuscripts utilizing custom algorithms or software that are central to the research but not yet described in published literature, software must be made available to editors and reviewers. We strongly encourage code deposition in a community repository (e.g. GitHub). See the Nature Portfolio guidelines for submitting code & software for further information.

## Data

Policy information about availability of data
All manuscripts must include a data availability statement. This statement should provide the following information, where applicable:
- Accession codes, unique identifiers, or web links for publicly available datasets
- A description of any restrictions on data availability
- For clinical datasets or third party data, please ensure that the statement adheres to our policy

> The data that support the findings of this study are included in this published article and its supplementary information. Structural data have been deposited in the PDB under the accession codes 9MD0 (tPBP235/02-12), 9MCZ (tPBP235/02-15) and 9Z5T (tPBP2H041-21). Raw sequence reads obtained from N. gonorrhoeae strains in this study were deposited in GenBank under BioProject accession number PRJNA1353147.

## Research involving human participants, their data, or biological material

Policy information about studies with human participants or human data. See also policy information about sex, gender (identity/presentation), and sexual orientation and race, ethnicity and racism.

| | |
|---|---|
| Reporting on sex and gender | N/A, neither human participants nor human data are reported in this manuscript. |
| Reporting on race, ethnicity, or other socially relevant groupings | N/A |
| Population characteristics | N/A |
| Recruitment | N/A |
| Ethics oversight | N/A |

Note that full information on the approval of the study protocol must also be provided in the manuscript.

# Field-specific reporting

Please select the one below that is the best fit for your research. If you are not sure, read the appropriate sections before making your selection.

☒ Life sciences ☐ Behavioural & social sciences ☐ Ecological, evolutionary & environmental sciences

For a reference copy of the document with all sections, see nature.com/documents/nr-reporting-summary-flat.pdf

# Life sciences study design

All studies must disclose on these points even when the disclosure is negative.

| | |
|---|---|
| Sample size | Murine PK studies (n=3 mice/group); Murine in vivo efficacy studies for boro-PBPi 18 (n=5 mice/group) and 21 (n=10 mice/group); Rat PK study (n=3 rats/group) |
| Data exclusions | No data were excluded from analyses. |
| Replication | The experiments were not replicated, because the sufficient samples of each group for examining significant difference were included in each study. |
| Randomization | Mice were randomized for drug/vehicle treatment. |
| Blinding | The investigators were not blinded as mice and samples were tracked from treatments and sampling to bioanalysis (colony counting or LC-MS/MS). |

# Reporting for specific materials, systems and methods

We require information from authors about some types of materials, experimental systems and methods used in many studies. Here, indicate whether each material, system or method listed is relevant to your study. If you are not sure if a list item applies to your research, read the appropriate section before selecting a response.

## Materials & experimental systems

| n/a | Involved in the study |
|---|---|
| ☒ | ☐ Antibodies |
| ☐ | ☒ Eukaryotic cell lines |
| ☒ | ☐ Palaeontology and archaeology |
| ☐ | ☒ Animals and other organisms |
| ☒ | ☐ Clinical data |
| ☒ | ☐ Dual use research of concern |
| ☒ | ☐ Plants |

## Methods

| n/a | Involved in the study |
|---|---|
| ☒ | ☐ ChIP-seq |
| ☒ | ☐ Flow cytometry |
| ☒ | ☐ MRI-based neuroimaging |

# Eukaryotic cell lines

Policy information about cell lines and Sex and Gender in Research

| | |
|---|---|
| Cell line source(s) | ATCC: MRC-5 (ATCC CCL-171), HeLa (ATCC CCL-2), 3T3 (ATCC CCL-92), SKOV-3 (ATCC HTB-77), CHO-K1 (ATCC CCL-61) |
| Authentication | None of the cell lines was authenticated. |
| Mycoplasma contamination | None of the cell lines was checked. |
| Commonly misidentified lines (See ICLAC register) | MRC-5, HeLa, and 3T3 are listed. The source of these cell lines used was ATCC as described above. |

# Animals and other research organisms

Policy information about studies involving animals; ARRIVE guidelines recommended for reporting animal research, and Sex and Gender in Research

| | |
|---|---|
| Laboratory animals | Female Balb/c mice for PK studies, 7–9 weeks old, received from Vital River, Zhejiang, China<br>Male CD-1 mice for PK studies, 7–9 weeks old, received from Vital River, Zhejiang, China<br>Female Balb/c mice for efficacy studies, 5–6 weeks old, received from BioLASCO Taiwan<br>Female NCI BALB/c mice for efficacy studies, 6–7 weeks old, received from Charles River Laboratories<br>Male Sprague Dawley rats for PK studies, 5–7 weeks old, received from Hilltop Labs Animals, Inc. |
| Wild animals | N/A |
| Reporting on sex | Female mice was used for in vivo efficacy studies because only vaginal infection models have been established to to test compound acitvity against N. gonorrhoeae. |
| Field-collected samples | N/A |
| Ethics oversight | Animal experiments were conducted at the Uniformed Services University (USU) of the Health Sciences (Bethesda, MD, USA), the BioDuro-Sundia DMPK group (Jiangsu, China), Eurofins Pharmacology Discovery Services (New Taipei City, Taiwan), and QPS, LLC (Newark, DE, USA). Each facility is accredited by the Association for the Assessment and Accreditation of Laboratory Animal Care (AAALAC) and is provided assurance from the National Institutes of Health Public Health Services' Office of Laboratory Animal Welfare (OLAW), with protocols approved by the Institutional Animal Care and Use Committees (IACUC). The IACUC-approved protocol numbers were BioDuro BD-202102114 (PK studies for boro-PBPi 18), BioDuro BDW-2201-0007 (PK studies for boro-PBPi 21), PDS IM005-07302018 (efficacy studies for boro-PBPi 18), USU MIC23-759 (efficacy studies for boro-PBPi 21), and QPS Number 005 (rat PK studies). |

Note that full information on the approval of the study protocol must also be provided in the manuscript.

# Plants

| | |
|---|---|
| Seed stocks | N/A |
| Novel plant genotypes | N/A |
| Authentication | N/A |

