## [Peer Review File · Nature Microbiology]

A penicillin-binding protein inhibitor series to target drug-resistant *Neisseria gonorrhoeae*

Corresponding Author: Dr Tsuyoshi Uehara

Version 0:

Reviewer comments:

Reviewer #1

(Remarks to the Author)

The authors have developed a series of novel benzoxaborinine-based penicillin-binding protein inhibitors (boro-PBPi) that bind to PBP2 and inhibit the growth of *N. gonorrhoeae*. Two compounds, 18 and 21, were tested for efficacy in the mouse vaginal colonization model using ovariectomized mice. Initially compound 18 at various doses was tested against lab strain FA1090 (sensitive to all antibiotics) to determine dosing guidelines. This was followed by testing compound 21 against CRO-resistant isolate H041 in the more commonly employed estradiol-treated mouse vaginal colonization model. The efficacy experiments in vivo have been clearly presented and the methods are robust. A couple of minor issues need clarification:

1. Can the authors clarify why initial testing done using ovariectomized mice and compound 18, while subsequent testing done in non-ovariectomized mice using compound 21?
2. In the efficacy experiment with compound 21, commensals (facultative aerobes) were monitored on HIA plates. Were there any obvious differences noted between the PBS groups and the antibiotic treated groups?

Reviewer #2

(Remarks to the Author)

The manuscript by Uehara et al. describes a novel class of benzoxaborinine-based PBP inhibitors (boro-PBPi) targeting drug resistant *Neisseria gonorrhoeae* including strains resistant to ceftriaxone (CRO). They developed and tested a series of compounds containing the benzoxaborinine core used in novel beta-lactamase inhibitors, with cephalosporin side chains. As benzoxaborinines are not beta-lactams they are not susceptible to beta-lactamases and are thus not expected to contribute to their expansion as a resistance mechanism. They identified compound 21, a boro-PBPi bearing the N-methylsulfonyl-2-imidazolinone motif from mezlocillin, which binds with high affinity to PBP2, including some mosaic variants. Compound 21 has potent activity against CRO-resistant *N. gonorrhoeae* bearing these PBPs as well as an extended spectrum beta-lactamase-producing strain. Structural analysis of compound 21 precursors bound to PBP2 revealed a covalent interaction of the boron atom with Ser310. The b3-b4 loop is moved toward the active site, suggesting flexibility, which is associated with enhanced affinity. This compound has favorable pharmacokinetic properties and safety profile in mice. The efficacy of compound 21 against CRO-resistant *N. gonorrhoeae* was demonstrated in a murine infection model. The study concludes that compound 21 is a promising anti-gonorrhea agent poised for preclinical and clinical development, offering a potential new therapeutic option for multidrug resistant *N. gonorrhoeae*. The manuscript is well-written and well-illustrated, the study is thorough and well-executed and the results are compelling. Below are a list of issues that should be addressed, ordered as they appear in the manuscript.

Line 151 – “A series of phenol-bearing boro-PBPi was prepared ...” sounds like a phenol was added when in fact it is already present but was modified. Clarify and state the rationale for modifying the phenol.

Line 190 – Explain why a phosphonate is preferable to a phenoxide or carboxylate.

Line 212-215 – The manuscript identifies compound 21 as the most efficacious, but presents structures of compounds 12 and 15 bound to PBP2. Were structures sought for compound 21 in complex with PBP2? Or mosaic PBP2? Compounds 12 and 15 differ from 21 in the side chain and phenol groups. Comment on where the side chains interact, how this might impact the b3-b4-loop and what these structures tell us about 21 binding. How might these boro-PBPis overcome the “conformational barrier created by resistance mutations”? (Line 236)

It would be helpful for non-chemists to know exactly which motifs are used in which compound. In at least some of the structure figures (eg. those in S2) indicate the benzoxaborinine core and the cephalosporin-based side chains/moieties.

Explain why compound 18 was tested on CRO-susceptible FA1090 in the in vivo mouse model whereas 21 was tested with CRO-sensitive H041? Why wasn't 21 tested for both *N. gonorrhoeae* strains? Why did the infection scheme differ for the two tests? What is the rationale for using ovariectomized BALB/c mice and what are the limitations? Why not use estradiol-treated mice with intact ovaries? (<https://pubmed.ncbi.nlm.nih.gov/31119637/>).

Line 281-284: "Notably, neither the evolution of spontaneous gonococcal mutants nor a significant change in vaginal polymorphonuclear leukocytes (PMNs) was observed between groups during this study (Figure S10)." Reference Fig. S11 as well.

No *N. gonorrhoeae* mutants were identified upon treatment with 4x and 16x compound 21 and ceftriaxone but it is well known that ceftriaxone mutants arise in natural infections. The authors should test for mutant evolution using sub-inhibitory concentrations to identify mutations that might emerge during treatment. Whole genome sequencing can be used to identify resistance mutations/targets.

Line 327 section – Provide context for each of the tests performed. Eg., define CYP enzymes and why these were selected for inhibition tests, why serine proteases, etc.

Line 341-391 – State here that compound 21 was administered subcutaneously in the in vivo efficacy test, in contrast to oral and intramuscular administration tested here.

Was compound 21 tested on *N. gonorrhoeae* biofilms? On bacterial species other than *N. gonorrhoeae*?

The Discussion needs work. Several of the above points should be expanded upon in this section (structures, mouse models, resistance) and the authors should outline next steps to bring this compound to clinical tests. This section ends rather abruptly and suggests that 21 will not be advanced further.

Reviewer #3

(Remarks to the Author)

In this manuscript by Uehara and colleagues, the development of a novel class of penicillin binding protein (PBP) inhibitors is described. Additionally, the authors demonstrate activity against *Neisseria gonorrhoeae*, including against strains with reduced susceptibility to the current recommended therapy ceftriaxone both in vitro and in the mouse model for gonorrhea. As *N. gonorrhoeae* has developed resistance to all previous recommended therapies, development of new antimicrobials targeting *N. gonorrhoeae* is of utmost importance. The manuscript is well written, and I found it enjoyable to read! As my own expertise is related to the genetic mechanisms underlying antimicrobial resistance in *N. gonorrhoeae*, my major comments on the manuscript are related to the potential for emergence of resistance to these novel compounds:

1. The authors found that among a panel of strains, two encoding mutations at codon 501 in *penA* had elevated MICs to compound 21. The authors suggest that resistance associated with these mutations are not of major concern because A501P has a demonstrated fitness cost and strains encoding this mutation are no longer transmitting. It may be worth mentioning here that while A501P is not commonly observed, A501V is present in currently circulating lineages (e.g. <https://www.microbiologyresearch.org/content/journal/mgen/10.1099/mgen.0.000480>).

2. Likewise, the authors demonstrated a low frequency of resistance for three *N. gonorrhoeae* strains, including WHO Q encoding *penA* 60. Do the authors expect that strains already encoding A501V or strains encoding *penA* 34 (where the A501P previously emerged) are more likely to evolve resistance to the novel compounds developed here?

Additional minor comments:

The authors use a strain panel including strains that are part of the WHO panel. However, some strains are referred to with WHO nomenclature and others with the original strain names. For example, F89 is WHO Y, and H041 is WHO X. Since WHO nomenclature is used for other strains in the study, I would suggest using WHO nomenclature throughout.

Line 135: As there are several mosaic PBP2 alleles that mediate reduced susceptibility to ceftriaxone, it would be helpful to list the *penA* allele encoded by H041/WHO X; particularly as *penA* 60 is mentioned in the introduction, and H041 does not encode *penA* 60. Likewise, it would be helpful to report the relevant *penA* alleles on Lines 206 and 215.

Line 148-149: Given the PBP2 binding data in Figure 1, it seems clear that variation associated with mosaic *penA* alleles contributes to activity of these compounds, but I wonder if the authors have considered whether mutation in *porB* (encoding the porin) or the *mtr* operon (encoding and efflux pump) are also contributing to MICs of the novel BLIs studied in this manuscript. FA19 and FA1090 are also WT at those loci, which contribute to resistance to multiple drugs.

Line 314/Table 2: This CDC strain panel is not particularly representative of the *N. gonorrhoeae* population as the majority of strains come from a single *penA* 34 encoding lineage of *N. gonorrhoeae*.

Line 324-325: It seems to me that it would be interesting to know the frequency of resistance for *penA* 34 encoding strains since 1) this lineage is still transmitting and 2) this lineage is only a single mutation from encoding *penA* 42.

Line 491: Have reads been deposited in a public repository?

Figure 1: There are a lot of colors associated with the table in this figure, and they are not described in the figure legend. In particular, the red/green color scheme for low/high PBP2 binding and MICs may not be ideal for readers with colorblindness.

Figure 3: It appears that the legend for panels B and C are mixed up.

Table S7: Is this genotyping from previously reported results? If not, are the methods used to identify variants reported in the methods?

Table S10: What do the colors in this table represent?

Reviewer #4

(Remarks to the Author)

This is an outstanding study that describes the discovery of a new class of penicillin-binding protein inhibitors designed to treat multidrug-resistant strains of *Neisseria gonorrhoea*, especially those strains that contain mosaic PBP2 mutations.

The design story describes fusing a boronic acid serine beta-lactamase warhead with Gram-negative cephalosporin sidechains to generate an initial PBP-targeted lead. This is followed by structure-guided rounds of lead optimization to ultimately generate VNRX-14079. It is an important antibiotic discovery story, well written and informative — just the sort of paper I would use in journal clubs for my trainees.

The study's high significance relates to the use of a non-beta-lactam scaffold to inhibit a PBP and, more specifically, generate potent compounds that can treat strains containing mosaic PBP2 mutations, which is no easy task. The lead's excellent drug-like properties include a low propensity for side effects and reasonable PK properties. Microbiological studies show activity against a broad panel of *Neisseria gonorrhoea* isolates, including those strains that contain mosaic PBP2 mutations. These results suggest that VNRX-14079 is a viable drug candidate for advancing towards IND status.

The supplemental data is highly supportive and contains much valuable information. Supporting the rigor and reproducibility of the study and the advancement of VNRX-14079.

The conclusion section mentions the real-world financial limitations of developing novel antibacterial agents. However, in this case, the authors have added to this problem by developing a non-oral drug for *Neisseria gonorrhoea*, which limits the potential market size.

Recommendations

1. The PBP2 binding assays are performed on isolated/truncated enzymes. Providing inhibition data from unbiased PBP profiling on 21 using a standard bocillin-PBP binding assay (See PMID: 39739989) on Ng isolates (mosaic and non-mosaic) would add important supportive data to this manuscript.
2. Please provide information on the route of elimination and primary metabolism of 21
3. Information on the microsomal and plasma stability data of 21 should be included in Table 3.
4. Could the authors comment on the safety of 21 as a poly-anion and its propensity to chelate metals? How may this affect the distribution to infected tissues? And how does the distribution of 21 compare to CRO?

Decision Letter:

23rd April 2025

Dear Professor Uehara,

Thank you for your patience while your manuscript "A new class of penicillin-binding protein inhibitors to address drug-resistant *Neisseria gonorrhoeae*" was under peer-review at Nature Microbiology. It has now been seen by 4 referees, whose expertise and comments you will find at the end of this email. Although they find your work of some potential interest, they have raised a number of concerns that will need to be addressed before we can consider publication of the work in Nature Microbiology.

In particular, referee #1 raises an important question of why two different mouse models were used, and why different compounds were tested. Referee #2 is concerned about why the structures are on compounds 12 and 15 instead of 21 (the latter

having been the most efficacious one). The referee says this should be discussed, especially what can be deduced about the structure of 21. Referee #2 also has concerns about the two different mouse models used and the different compounds tested. Further, referee #2 asks for further testing of resistance evolution, and for testing 21 against biofilms and other bacterial species. Referee #3 has some questions regarding resistance evolution. Referee #4 asks to provide on Ng isolates (mosaic and non-mosaic). The referee also asks to provide information on the route of elimination and primary metabolism of 21, and information on the microsomal and plasma stability data of 21. Editorially, we will require all referee comments to be addressed in full, and the main points are listed below:

1. Perform structural analysis of 21, or if not possible, better describe what we can learn about 21 based on the two other structures
2. Ideally, perform the in vivo mouse models using the same model and compound (or explain if there is a misunderstanding here)
3. Test for mutant evolution using sub-inhibitory concentrations
4. Test 21 against at least 2-3 other bacterial species

Should further experimental data allow you to address these criticisms, we would be happy to look at a revised manuscript.

Please include a data availability statement as a separate section after Methods but before references, under the heading "Data Availability". This section should inform readers about the availability of the data used to support the conclusions of your study. This information includes accession codes to public repositories (data banks for protein, DNA or RNA sequences, microarray, proteomics data etc...), references to source data published alongside the paper, unique identifiers such as URLs to data repository entries, or data set DOIs, and any other statement about data availability. At a minimum, you should include the following statement: "The data that support the findings of this study are available from the corresponding author upon request", mentioning any restrictions on availability. If DOIs are provided, we also strongly encourage including these in the Reference list (authors, title, publisher (repository name), identifier, year). For more guidance on how to write this section please see: <http://www.nature.com/authors/policies/data/data-availability-statements-data-citations.pdf>

* If you have not done so already we suggest that you begin to revise your manuscript so that it conforms to our Article format instructions at <http://www.nature.com/nmicrobiol/info/final-submission>. Refer also to any guidelines provided in this letter.

When submitting the revised version of your manuscript, please pay close attention to our [href="https://www.nature.com/nature-portfolio/editorial-policies/image-integrity">Digital Image Integrity Guidelines. and to the following points below:](https://www.nature.com/nature-portfolio/editorial-policies/image-integrity)

- that unprocessed scans are clearly labelled and match the gels and western blots presented in figures.
- that control panels for gels and western blots are appropriately described as loading or sample processing controls
- all images in the paper are checked for duplication of panels and for splicing of gel lanes.

EXTENDED DATA FIGURES

Link Redacted

Note: This url links to your confidential homepage and associated information about manuscripts you may have submitted or be reviewing for us. If you wish to forward this e-mail to co-authors, please delete this link to your homepage first.

Nature Microbiology is committed to improving transparency in authorship. As part of our efforts in this direction, we are now requesting that all authors identified as 'corresponding author' on published papers create and link their Open Researcher and Contributor Identifier (ORCID) with their account on the Manuscript Tracking System (MTS), prior to acceptance. This applies to primary research papers only. ORCID helps the scientific community achieve unambiguous attribution of all scholarly contributions. You can create and link your ORCID from the home page of the MTS by clicking on 'Modify my Springer Nature account'. For more information please visit www.springernature.com/orcid.

If you wish to submit a suitably revised manuscript we would hope to receive it within 6 months. If you cannot send it within this time, please let us know.

Yours sincerely,

Reviewer Expertise:

Referee #1: In vivo mouse model
Referee #2: Structural biology
Referee #3: Neisseria, AMR
Referee #4: Antimicrobials, chemistry

Reviewer Comments:

Reviewer #1 (Remarks to the Author):

The authors have developed a series of novel benzoxaborinine-based penicillin-binding protein inhibitors (boro-PBPi) that bind to PBP2 and inhibit the growth of *N. gonorrhoeae*. Two compounds, 18 and 21, were tested for efficacy in the mouse vaginal colonization model using ovariectomized mice. Initially compound 18 at various doses was tested against lab strain FA1090 (sensitive to all antibiotics) to determine dosing guidelines. This was followed by testing compound 21 against CRO-resistant isolate H041 in the more commonly employed estradiol-treated mouse vaginal colonization model. The efficacy experiments in vivo have been clearly presented and the methods are robust. A couple of minor issues need clarification:

1. Can the authors clarify why initial testing done using ovariectomized mice and compound 18, while subsequent testing done in non-ovariectomized mice using compound 21?
2. In the efficacy experiment with compound 21, commensals (facultative aerobes) were monitored on HIA plates. Were there any obvious differences noted between the PBS groups and the antibiotic treated groups?

Reviewer #2 (Remarks to the Author):

The manuscript by Uehara et al. describes a novel class of benzoxaborinine-based PBP inhibitors (boro-PBPi) targeting drug resistant *Neisseria gonorrhoeae* including strains resistant to ceftriaxone (CRO). They developed and tested a series of compounds containing the benzoxaborinine core used in novel beta-lactamase inhibitors, with cephalosporin side chains. As benzoxaborinines are not beta-lactams they are not susceptible to beta-lactamases and are thus not expected to contribute to their expansion as a resistance mechanism. They identified compound 21, a boro-PBPi bearing the N-methylsulfonyl-2-imidazolinone motif from mezlocillin, which binds with high affinity to PBP2, including some mosaic variants. Compound 21 has potent activity against CRO-resistant *N. gonorrhoeae* bearing these PBPs as well as an extended spectrum beta-lactamase-producing strain. Structural analysis of compound 21 precursors bound to PBP2 revealed a covalent interaction of the boron atom with Ser310. The b3-b4 loop is moved toward the active site, suggesting flexibility, which is associated with enhanced affinity. This compound has favorable pharmacokinetic properties and safety profile in mice. The efficacy of compound 21 against CRO-resistant *N. gonorrhoeae* was demonstrated in a murine infection model. The study concludes that compound 21 is a promising anti-gonorrhoea agent poised for preclinical and clinical development, offering a potential new therapeutic option for multidrug resistant *N. gonorrhoeae*. The manuscript is well-written and well-illustrated, the study is thorough and well-executed and the results are compelling. Below are a list of issues that should be addressed, ordered as they appear in the manuscript.

Line 151 – "A series of phenol-bearing boro-PBPi was prepared ..." sounds like a phenol was added when in fact it is already present but was modified. Clarify and state the rationale for modifying the phenol.

Line 190 – Explain why a phosphonate is preferable to a phenoxide or carboxylate.

Line 212-215 – The manuscript identifies compound 21 as the most efficacious, but presents structures of compounds 12 and 15

bound to PBP2. Were structures sought for compound 21 in complex with PBP2? Or mosaic PBP2? Compounds 12 and 15 differ from 21 in the side chain and phenol groups. Comment on where the side chains interact, how this might impact the b3-b4-loop and what these structures tell us about 21 binding. How might these boro-PBPis overcome the “conformational barrier created by resistance mutations”? (Line 236)

It would be helpful for non-chemists to know exactly which motifs are used in which compound. In at least some of the structure figures (eg. those in S2) indicate the benzoxaborinine core and the cephalosporin-based side chains/moieties.

Explain why compound 18 was tested on CRO-susceptible FA1090 in the in vivo mouse model whereas 21 was tested with CRO-sensitive H041? Why wasn't 21 tested for both *N. gonorrhoeae* strains? Why did the infection scheme differ for the two tests? What is the rationale for using ovariectomized BALB/c mice and what are the limitations? Why not use estradiol-treated mice with intact ovaries? (<https://pubmed.ncbi.nlm.nih.gov/31119637/>).

Line 281-284: “Notably, neither the evolution of spontaneous gonococcal mutants nor a significant change in vaginal polymorphonuclear leukocytes (PMNs) was observed between groups during this study (Figure S10).” Reference Fig. S11 as well.

No *N. gonorrhoeae* mutants were identified upon treatment with 4x and 16x compound 21 and ceftriaxone but it is well known that ceftriaxone mutants arise in natural infections. The authors should test for mutant evolution using sub-inhibitory concentrations to identify mutations that might emerge during treatment. Whole genome sequencing can be used to identify resistance mutations/targets.

Line 327 section – Provide context for each of the tests performed. Eg., define CYP enzymes and why these were selected for inhibition tests, why serine proteases, etc.

Line 341-391 – State here that compound 21 was administered subcutaneously in the in vivo efficacy test, in contrast to oral and intramuscular administration tested here.

Was compound 21 tested on *N. gonorrhoeae* biofilms? On bacterial species other than *N. gonorrhoeae*?

The Discussion needs work. Several of the above points should be expanded upon in this section (structures, mouse models, resistance) and the authors should outline next steps to bring this compound to clinical tests. This section ends rather abruptly and suggests that 21 will not be advanced further.

Reviewer #3 (Remarks to the Author):

In this manuscript by Uehara and colleagues, the development of a novel class of penicillin binding protein (PBP) inhibitors is described. Additionally, the authors demonstrate activity against *Neisseria gonorrhoeae*, including against strains with reduced susceptibility to the current recommended therapy ceftriaxone both in vitro and in the mouse model for gonorrhea. As *N. gonorrhoeae* has developed resistance to all previous recommended therapies, development of new antimicrobials targeting *N. gonorrhoeae* is of utmost importance. The manuscript is well written, and I found it enjoyable to read! As my own expertise is related to the genetic mechanisms underlying antimicrobial resistance in *N. gonorrhoeae*, my major comments on the manuscript are related to the potential for emergence of resistance to these novel compounds:

1. The authors found that among a panel of strains, two encoding mutations at codon 501 in *penA* had elevated MICs to compound 21. The authors suggest that resistance associated with these mutations are not of major concern because A501P has a demonstrated fitness cost and strains encoding this mutation are no longer transmitting. It may be worth mentioning here that while A501P is not commonly observed, A501V is present in currently circulating lineages (e.g. <https://www.microbiologyresearch.org/content/journal/mgen/10.1099/mgen.0.000480>).

2. Likewise, the authors demonstrated a low frequency of resistance for three *N. gonorrhoeae* strains, including WHO Q encoding *penA* 60. Do the authors expect that strains already encoding A501V or strains encoding *penA* 34 (where the A501P previously emerged) are more likely to evolve resistance to the novel compounds developed here?

Additional minor comments:

The authors use a strain panel including strains that are part of the WHO panel. However, some strains are referred to with WHO nomenclature and others with the original strain names. For example, F89 is WHO Y, and H041 is WHO X. Since WHO nomenclature is used for other strains in the study, I would suggest using WHO nomenclature throughout.

Line 135: As there are several mosaic PBP2 alleles that mediate reduced susceptibility to ceftriaxone, it would be helpful to list the *penA* allele encoded by H041/WHO X; particularly as *penA* 60 is mentioned in the introduction, and H041 does not encode *penA* 60. Likewise, it would be helpful to report the relevant *penA* alleles on Lines 206 and 215.

Line 148-149: Given the PBP2 binding data in Figure 1, it seems clear that variation associated with mosaic *penA* alleles contributes to activity of these compounds, but I wonder if the authors have considered whether mutation in *porB* (encoding the porin) or the *mtr* operon (encoding and efflux pump) are also contributing to MICs of the novel BLIs studied in this manuscript. FA19 and FA1090 are also WT at those loci, which contribute to resistance to multiple drugs.

Line 314/Table 2: This CDC strain panel is not particularly representative of the *N. gonorrhoeae* population as the majority of strains come from a single penA 34 encoding lineage of *N. gonorrhoeae*.

Line 324-325: It seems to me that it would be interesting to know the frequency of resistance for penA 34 encoding strains since 1) this lineage is still transmitting and 2) this lineage is only a single mutation from encoding penA 42.

Line 491: Have reads been deposited in a public repository?

Figure 1: There are a lot of colors associated with the table in this figure, and they are not described in the figure legend. In particular, the red/green color scheme for low/high PBP2 binding and MICs may not be ideal for readers with colorblindness.

Figure 3: It appears that the legend for panels B and C are mixed up.

Table S7: Is this genotyping from previously reported results? If not, are the methods used to identify variants reported in the methods?

Table S10: What do the colors in this table represent?

Reviewer #4 (Remarks to the Author):

This is an outstanding study that describes the discovery of a new class of penicillin-binding protein inhibitors designed to treat multidrug-resistant strains of *Neisseria gonorrhoea*, especially those strains that contain mosaic PBP2 mutations.

The design story describes fusing a boronic acid serine beta-lactamase warhead with Gram-negative cephalosporin sidechains to generate an initial PBP-targeted lead. This is followed by structure-guided rounds of lead optimization to ultimately generate VNRX-14079. It is an important antibiotic discovery story, well written and informative — just the sort of paper I would use in journal clubs for my trainees.

The study's high significance relates to the use of a non-beta-lactam scaffold to inhibit a PBP and, more specifically, generate potent compounds that can treat strains containing mosaic PBP2 mutations, which is no easy task. The lead's excellent drug-like properties include a low propensity for side effects and reasonable PK properties. Microbiological studies show activity against a broad panel of *Neisseria gonorrhoea* isolates, including those strains that contain mosaic PBP2 mutations. These results suggest that VNRX-14079 is a viable drug candidate for advancing towards IND status.

The supplemental data is highly supportive and contains much valuable information. Supporting the rigor and reproducibility of the study and the advancement of VNRX-14079.

The conclusion section mentions the real-world financial limitations of developing novel antibacterial agents. However, in this case, the authors have added to this problem by developing a non-oral drug for *Neisseria gonorrhoea*, which limits the potential market size.

Recommendations

1. The PBP2 binding assays are performed on isolated/truncated enzymes. Providing inhibition data from unbiased PBP profiling on 21 using a standard bocillin-PBP binding assay (See PMID: 39739989) on Ng isolates (mosaic and non-mosaic) would add important supportive data to this manuscript.
2. Please provide information on the route of elimination and primary metabolism of 21
3. Information on the microsomal and plasma stability data of 21 should be included in Table 3.
4. Could the authors comment on the safety of 21 as a poly-anion and its propensity to chelate metals? How may this affect the distribution to infected tissues? And how does the distribution of 21 compare to CRO?

Version 1:

Reviewer comments:

Reviewer #1

(Remarks to the Author)

The authors have responded to my critiques satisfactorily. I have no further comments.

I congratulate the authors on this fine piece of science. It is most unfortunate this program has been terminated because of a lack of interest from private funders. Such decisions will worsen the AMR crisis.

Reviewer #2

(Remarks to the Author)

The authors have addressed my concerns in the revised manuscript. This is an impressive and exhaustive study.

Lisa Craig

Reviewer #3

(Remarks to the Author)

In this revision, the authors have addressed my previous comments, including performing experiments on resistance emergence in WHO L and CDC-0197 strains after exposure to 4X, 16X, and sub-MIC concentrations of compound 21 and using isogenic mutants to explore the role of the porin and mtr efflux pump in resistance to compound 21.

I can also confirm that all the sequencing data associated with these experiments is available in NCBI's SRA. Since the mtr pump is the major antibiotic efflux pump in gonococcus, I was quite surprised by the premature stop codon in mtrE that was found in passage 8 of the WHO L experiment. I looked at the data myself and found the same mutations as the authors!

I have a couple of minor comments on the revision:

Line 349: Is the word efflux missing?

Line 487-489: This blast search is mentioned for the first time in the discussion section. I would suggest putting this in the results section and mentioning this analysis in the methods as well.

Extended Data Figure 7: It would be helpful if protein names used in the gonococcal literature were used here like in the main text (e.g. Opa proteins, MtrE).

Reviewer #4

(Remarks to the Author)

The authors have done an excellent job responding to the review, and I have no further suggestions for revisions. This is an outstanding antibacterial drug discovery paper that deserves publication.

Decision Letter:

Our ref: NMICROBIOL-24124046A

19th December 2025

Dear Tsuyoshi,

Thank you for submitting your revised manuscript "A new class of penicillin-binding protein inhibitors to address drug-resistant *Neisseria gonorrhoeae*" (NMICROBIOL-24124046A). It has now been seen by the original referees and their comments are below. The reviewers find that the paper has improved in revision, and therefore we'll be happy in principle to publish it in Nature Microbiology, pending minor revisions to satisfy the referees' final requests and to comply with our editorial and formatting guidelines.

We are now performing detailed checks on your paper and will send you a checklist detailing our editorial and formatting requirements in about 2-3 weeks (our office will be closed from Dec 22 - Jan 2nd). Please do not upload the final materials and make any revisions until you receive this additional information from us.

Thank you again for your interest in Nature Microbiology. Please do not hesitate to contact me if you have any questions.

Sincerely,

Reviewer #1 (Remarks to the Author):

The authors have responded to my critiques satisfactorily. I have no further comments.

I congratulate the authors on this fine piece of science. It is most unfortunate this program has been terminated because of a lack

of interest from private funders. Such decisions will worsen the AMR crisis.

Reviewer #2 (Remarks to the Author):

The authors have addressed my concerns in the revised manuscript. This is an impressive and exhaustive study.

Lisa Craig

Reviewer #3 (Remarks to the Author):

In this revision, the authors have addressed my previous comments, including performing experiments on resistance emergence in WHO L and CDC-0197 strains after exposure to 4X, 16X, and sub-MIC concentrations of compound 21 and using isogenic mutants to explore the role of the porin and mtr efflux pump in resistance to compound 21.

I can also confirm that all the sequencing data associated with these experiments is available in NCBI's SRA. Since the mtr pump is the major antibiotic efflux pump in gonococcus, I was quite surprised by the premature stop codon in mtrE that was found in passage 8 of the WHO L experiment. I looked at the data myself and found the same mutations as the authors!

I have a couple of minor comments on the revision:

Line 349: Is the word efflux missing?

Line 487-489: This blast search is mentioned for the first time in the discussion section. I would suggest putting this in the results section and mentioning this analysis in the methods as well.

Extended Data Figure 7: It would be helpful if protein names used in the gonococcal literature were used here like in the main text (e.g. Opa proteins, MtrE).

Reviewer #4 (Remarks to the Author):

The authors have done an excellent job responding to the review, and I have no further suggestions for revisions. This is an outstanding antibacterial drug discovery paper that deserves publication.

Version 2:

Decision Letter:

26th February 2026

Dear Tsuyoshi,

I am pleased to accept your Article "A penicillin-binding protein inhibitor series to target drug-resistant *Neisseria gonorrhoeae*" for publication in Nature Microbiology. Thank you for having chosen to submit your work to us and many congratulations.

Authors may need to take specific actions to achieve compliance with funder and institutional open access mandates. If your research is supported by a funder that requires immediate open access (e.g. according to [Plan S principles](https://www.springernature.com/gp/open-science/plan-s-compliance) or the [NIH public access policy](https://www.springernature.com/gp/open-science/us-federal-agency-compliance)) then you should select the gold OA route, and we will direct you to the compliant route where possible. Because authors warrant under our subscription licensing terms that they haven't committed to licensing any version of their article under a licence inconsistent with the terms of our agreement – including the applicable embargo period – publication under the subscription model isn't suitable for authors whose funders require no embargo.

Congratulations once again and I look forward to seeing the article published.

With kind regards,

P.S. Click on the following link if you would like to recommend Nature Microbiology to your librarian <http://www.nature.com/subscriptions/recommend.html#forms>

** Visit the Springer Nature Editorial and Publishing website at http://editorial-jobs.springernature.com?utm_source=ejp_NMicro_email&utm_medium=ejp_NMicro_email&utm_campaign=ejp_NMicro for more information about our career opportunities. If you have any questions please click [here](mailto:editorial.publishing.jobs@springernature.com).**

Response to Reviewers' Comments (author response shown in purple)

Reviewer #1 (Remarks to the Author):

The authors have developed a series of novel benzoxaborinine-based penicillin-binding protein inhibitors (boro-PBPi) that bind to PBP2 and inhibit the growth of *N. gonorrhoeae*. Two compounds, 18 and 21, were tested for efficacy in the mouse vaginal colonization model using ovariectomized mice. Initially compound 18 at various doses was tested against lab strain FA1090 (sensitive to all antibiotics) to determine dosing guidelines. This was followed by testing compound 21 against CRO-resistant isolate H041 in the more commonly employed estradiol-treated mouse vaginal colonization model. The efficacy experiments in vivo have been clearly presented and the methods are robust. A couple of minor issues need clarification:

1. Can the authors clarify why initial testing done using ovariectomized mice and compound 18, while subsequent testing done in non-ovariectomized mice using compound 21?

The in vivo model with the susceptible FA1090 strain (used for compound **18**) was used as a proof of concept to show that boro-PBPi has in vivo efficacy against *N. gonorrhoeae*. In addition, practicality was another factor because this infection model was available at Eurofins through CARB-X's financial support and NIAID Pre-Clinical Services, while it was not possible to perform the in vivo study in Ann Jerse's lab in a timely manner due to operational challenges encountered during the COVID-19 pandemic. The in vivo model with the ceftriaxone-resistant H041 strain (used for compound **21**) was performed later by Ann Jerse's group to show the efficacy of boro-PBPi against this challenging strain. In the revised manuscript, we have added statements (lines 296-300 and 310-313) to clarify why the different infection models were used for compounds **18** and **21**.

2. In the efficacy experiment with compound 21, commensals (facultative aerobes) were monitored on HIA plates. Were there any obvious differences noted between the PBS groups and the antibiotic treated groups?

In the untreated and CRO-treated controls, as well as the test groups that received 10 mg/kg q12h, 200 mg/kg q24h, and 150 mg/kg q8h, 0-20% of mice had bacterial growth on HIA plates during the study. The majority of isolates were Gram-positive cocci in clusters, and some isolates were Gram-positive diplococci, probably *Staphylococcus* sp. and *Enterococcus* sp., respectively. In the test group (150 mg/kg, q12h), 60% of the mice were colonized with *Staphylococcus*-like bacteria. *Staphylococcal* presence in vaginal microbiota does not usually affect the duration of gonococcal colonization or the bioburden. Statements regarding this observation have been added in the revised Supplementary Information (added in the **Table S4** legend).

Reviewer #2 (Remarks to the Author):

The manuscript by Uehara et al. describes a novel class of benzoxaborinine-based PBP inhibitors (boro-PBPi) targeting drug resistant *Neisseria gonorrhoeae* including strains resistant to ceftriaxone (CRO). They developed and tested a series of compounds containing the benzoxaborinine core used in novel beta-lactamase inhibitors, with cephalosporin side chains. As benzoxaborinines are not beta-lactams they are not susceptible to beta-lactamases and are thus not expected to contribute to their expansion as a resistance mechanism. They identified

compound 21, a boro-PBPi bearing the N-methylsulfonyl-2-imidazolinone motif from mezlocillin, which binds with high affinity to PBP2, including some mosaic variants. Compound 21 has potent activity against CRO-resistant *N. gonorrhoeae* bearing these PBPs as well as an extended spectrum beta-lactamase-producing strain. Structural analysis of compound 21 precursors bound to PBP2 revealed a covalent interaction of the boron atom with Ser310. The b3-b4 loop is moved toward the active site, suggesting flexibility, which is associated with enhanced affinity. This compound has favorable pharmacokinetic properties and safety profile in mice. The efficacy of compound 21 against CRO-resistant *N. gonorrhoeae* was demonstrated in a murine infection model. The study concludes that compound 21 is a promising anti-gonorrhea agent poised for preclinical and clinical development, offering a potential new therapeutic option for multidrug resistant *N. gonorrhoeae*. The manuscript is well-written and well-illustrated, the study is thorough and well-executed and the results are compelling. Below are a list of issues that should be addressed, ordered as they appear in the manuscript.

Line 151 – “A series of phenol-bearing boro-PBPi was prepared ...” sounds like a phenol was added when in fact it is already present but was modified. Clarify and state the rationale for modifying the phenol.

A rationale has been added for clarification (lines 152-153).

Line 190 – Explain why a phosphonate is preferable to a phenoxide or carboxylate.

A statement has been added into the revised manuscript (lines 197-200) to clarify why the phosphonate substitution was examined. The superior activity of the phosphonate compound can be explained by the binding modes observed in the obtained PBP2/boro-PBPi complex structures (carboxylate boro-PBPi **12** vs. phosphonate **15**), which is described in the “Results” section “Structural evidence for high affinity binding of boro-PBPi to mosaic PBP2”.

Line 212-215 – The manuscript identifies compound 21 as the most efficacious, but presents structures of compounds 12 and 15 bound to PBP2. Were structures sought for compound 21 in complex with PBP2? Or mosaic PBP2? Compounds 12 and 15 differ from 21 in the side chain and phenol groups. Comment on where the side chains interact, how this might impact the b3-b4-loop and what these structures tell us about 21 binding. How might these boro-PBPis overcome the “conformational barrier created by resistance mutations”? (Line 236)

Since the original submission, we have now obtained a crystal structure of mosaic PBP2^{H041} in complex with 21. Its phosphonate forms similar interactions involving His514, Tyr543 and Arg503 as the phosphonate in **15**. Also similar to the complexes with **12** and **15**, the β_3 - β_4 loop occupies an inward position commensurate with higher-activity inhibitors of PBP2.

The new structure has been added into the revised manuscript and **Figure 2** accordingly.

It would be helpful for non-chemists to know exactly which motifs are used in which compound. In at least some of the structure figures (eg. those in S2) indicate the benzoxaborinine core and the cephalosporin-based side chains/moieties.

In **Figures S1** and **S2**, the benzoxaborinine core and the β -lactam side chains have been color-labeled.

Explain why compound 18 was tested on CRO-susceptible FA1090 in the in vivo mouse model whereas 21 was tested with CRO-sensitive H041? Why wasn't 21 tested for both *N. gonorrhoeae* strains? Why did the infection scheme differ for the two tests? What is the

rationale for using ovariectomized BALB/c mice and what are the limitations? Why not use estradiol-treated mice with intact ovaries? (<https://pubmed.ncbi.nlm.nih.gov/31119637/>).

This in vivo model using susceptible FA1090 was used as a proof of concept to show that boro-PBPi has in vivo efficacy against *N. gonorrhoeae*. In addition, practicality was another factor because this infection model was available at Eurofins through CARB-X's financial support and NIAID Pre-Clinical Services, while it was not possible to perform the in vivo study in Ann Jerse's lab in a timely manner due to operational challenges encountered during the COVID-19 pandemic. In the revised manuscript, we have added statements to explain this (lines 296-300 and 310-313). The only murine infection model available at Eurofins was using ovariectomized BALB/c mice to establish colonization of *N. gonorrhoeae* FA1090.

Line 281-284: "Notably, neither the evolution of spontaneous gonococcal mutants nor a significant change in vaginal polymorphonuclear leukocytes (PMNs) was observed between groups during this study (Figure S10)." Reference Fig. S11 as well.

We appreciate your comments. **Fig. S11** of the original manuscript (**Extended Data Figure 6** in the revised manuscript) shows in vitro bactericidal activity not the in vivo study, so we did not add this figure as a reference, as suggested. Instead, we have added **Table S4 and S5** (line 328) as references, because these show the raw data of CFU and PMN measured in the in vivo study.

No *N. gonorrhoeae* mutants were identified upon treatment with 4x and 16x compound 21 and ceftriaxone but it is well known that ceftriaxone mutants arise in natural infections. The authors should test for mutant evolution using sub-inhibitory concentrations to identify mutations that might emerge during treatment. Whole genome sequencing can be used to identify resistance mutations/targets.

We performed 10-day passaging experiments with sub-MIC of compound **21** using ATCC 49226, WHO L, CDC-0197, and WHO Q strains. We obtained reduced susceptibility mutants from WHO L and CDC-0197 at day 8. These results have been added in the revised manuscript (lines 382-395 and **Extended Data Figure 7**).

Line 327 section – Provide context for each of the tests performed. Eg., define CYP enzymes and why these were selected for inhibition tests, why serine proteases, etc.

Clarification notes for these selectivity tests have been added in the revised manuscript (lines 402-410).

Line 341-391 – State here that compound 21 was administered subcutaneously in the in vivo efficacy test, in contrast to oral and intramuscular administration tested here.

We used subcutaneous administration in the in vivo efficacy studies because intramuscular administration in mice is challenging owing to the small injection volume allowed (0.1 mL/day) compared to the acceptable volume for SC administration (10 mL/kg/site/day) [reference PMID: 11180276]. We have added this information into the revised manuscript (lines 419-421).

The oral absorption of compound **21** (and other similar boro-PBPi) was expected to be poor in humans based on the Caco-2 data. Therefore, we did not use oral administration of boro-PBPi. Through our experience in discovery and development of the oral boron-based beta-lactamase inhibitor VNRX-7145 (i.e., ledaborbactam), we have found that ideal oral absorption of benzoxaborinine-containing compounds requires the use of prodrug forms of acidic moieties as

well as low molecular weight for such prodrugs, which was challenging to accomplish for these boro-PBPi.

Was compound 21 tested on *N. gonorrhoeae* biofilms? On bacterial species other than *N. gonorrhoeae*?

We did not test the activity of **21** against *N. gonorrhoeae* biofilms. A study on biofilms is a future plan, as stated in the revised manuscript (line 496).

The activity of 21 against other bacterial species other than *N. gonorrhoeae* is shown in **Extended Data Table 1 (Table S1)** of the original manuscript).

The Discussion needs work. Several of the above points should be expanded upon in this section (structures, mouse models, resistance) and the authors should outline next steps to bring this compound to clinical tests. This section ends rather abruptly and suggests that 21 will not be advanced further.

As suggested, the Discussion section of the revised manuscript has been expanded on crystal structures, mouse models, resistance, and future steps. Additionally, we have revised the last paragraph of Discussion (lines 491-500), replacing the previous commentary on the current real-world situation of the antibiotic industry with next steps to advance compound **21** as an anti-gonorrhea agent. However, it is pertinent to highlight the current challenging business environment surrounding antibiotic discovery and development. For instance, in mid-2025, Venatorx terminated all discovery and development activities, resulting in the layoff of nearly all employees, including all Venatorx authors of this manuscript. The discovery of **21** failed to attract interest from private funding sources, such as venture capital, primarily due to the lack of anticipated profitability in developing an anti-gonorrhoeae drug within the current broken antibiotic market.

Reviewer #3 (Remarks to the Author):

In this manuscript by Uehara and colleagues, the development of a novel class of penicillin binding protein (PBP) inhibitors is described. Additionally, the authors demonstrate activity against *Neisseria gonorrhoeae*, including against strains with reduced susceptibility to the current recommended therapy ceftriaxone both in vitro and in the mouse model for gonorrhea. As *N. gonorrhoeae* has developed resistance to all previous recommended therapies, development of new antimicrobials targeting *N. gonorrhoeae* is of utmost importance. The manuscript is well written, and I found it enjoyable to read! As my own expertise is related to the genetic mechanisms underlying antimicrobial resistance in *N. gonorrhoeae*, my major comments on the manuscript are related to the potential for emergence of resistance to these novel compounds:

1. The authors found that among a panel of strains, two encoding mutations at codon 501 in *penA* had elevated MICs to compound 21. The authors suggest that resistance associated with these mutations are not of major concern because A501P has a demonstrated fitness cost and strains encoding this mutation are no longer transmitting. It may be worth mentioning here that while A501P is not commonly observed, A501V is present in currently circulating lineages (e.g. <https://www.microbiologyresearch.org/content/journal/mgen/10.1099/mgen.0.000480>).

A statement describing the presence of A501V in currently circulating lineages has been added in the revised manuscript, as suggested (lines 345-347).

2. Likewise, the authors demonstrated a low frequency of resistance for three *N. gonorrhoeae* strains, including WHO Q encoding *penA* 60. Do the authors expect that strains already encoding A501V or strains encoding *penA* 34 (where the A501P previously emerged) are more likely to evolve resistance to the novel compounds developed here?

We performed FoR experiments WHO L and CDC-0197(*penA*34). The results have been added into the revised manuscript (lines 377-380 and **Table S10**).

Additional minor comments:

The authors use a strain panel including strains that are part of the WHO panel. However, some strains are referred to with WHO nomenclature and others with the original strain names. For example, F89 is WHO Y, and H041 is WHO X. Since WHO nomenclature is used for other strains in the study, I would suggest using WHO nomenclature throughout.

The names H041 and F89 have been used in many publications, whereas the corresponding WHO strain names (WHO X and Y, respectively) have not. Other WHO strains do not have general strain names other than the WHO names. Thus, we retained the strain names as they are. However, in the revised manuscript, the WHO strain designations have been incorporated at the initial mention of H041 and F89 (lines 135 and 337, respectively).

Line 135: As there are several mosaic PBP2 alleles that mediate reduced susceptibility to ceftriaxone, it would be helpful to list the *penA* allele encoded by H041/WHO X; particularly as *penA* 60 is mentioned in the introduction, and H041 does not encode *penA* 60. Likewise, it would be helpful to report the relevant *penA* alleles on Lines 206 and 215.

The corresponding *penA* alleles have been added in the revised manuscript (lines 135-136).

Line 148-149: Given the PBP2 binding data in Figure 1, it seems clear that variation associated with mosaic *penA* alleles contributes to activity of these compounds, but I wonder if the authors have considered whether mutation in *porB* (encoding the porin) or the *mtr* operon (encoding and efflux pump) are also contributing to MICs of the novel BLIs studied in this manuscript. FA19 and FA1090 are also WT at those loci, which contribute to resistance to multiple drugs.

In the revised manuscript, we have added MIC data against isogenic strains to address the effect of the PorB porin and Mtr efflux system on boro-PBPi (**Table S8**). Minimal effect of the PorB porin and the Mtr efflux was observed on boro-PBPi activity (lines 348-350 in the revised manuscript).

Line 314/Table 2: This CDC strain panel is not particularly representative of the *N. gonorrhoeae* population as the majority of strains come from a single *penA* 34 encoding lineage of *N. gonorrhoeae*.

The CDC strain panel was formed from surveillance studies conducted in the US, not based on genetic diversity. The *penA* information is not described in the main text reference 61 or a public information. Our genome sequencing analysis revealed for the first time that most of these CDC strains carried *penA*-34. The sentence has been modified to clarify the points in the revised manuscript (lines 366-368).

The activity against the WHO reference strain panel listed in **Table 1** reflects the activity of boron-based PBP inhibitors against more diverse strains than that of the CDC strain panel.

Line 324-325: It seems to me that it would be interesting to know the frequency of resistance for penA 34 encoding strains since 1) this lineage is still transmitting and 2) this lineage is only a single mutation from encoding penA 42.

As described above, FoR studies with WHO L and CDC-0197 were performed, and the data have been added in **Table S10** (4x MIC and 16x MIC) in the revised manuscript (lines 377-380). We did not obtain any colonies from either strain, and the FoR was $<5 \times 10^{-9}$ against these strains.

Line 491: Have reads been deposited in a public repository?

WGS reads were deposited in the NCBI GenBank with the BioProject ID: PRJNA1353147.

Figure 1: There are a lot of colors associated with the table in this figure, and they are not described in the figure legend. In particular, the red/green color scheme for low/high PBP2 binding and MICs may not be ideal for readers with colorblindness.

As suggested, the table in **Figure 1** has been updated in the revised manuscript.

Figure 3: It appears that the legend for panels B and C are mixed up.

Thank you. The panels B and C have been swapped to align with the legend.

Table S7: Is this genotyping from previously reported results? If not, are the methods used to identify variants reported in the methods?

The genotypes of these strains have been previously described (references 73 and 74 in the main text). To confirm that the strains used were correct, we performed and analyzed genome sequencing. The description of the methods used for the WGS analysis and **Table S13** have been added in the revised manuscript, and the raw sequence data have been deposited in the GenBank database (PRJNA1353147).

Table S10: What do the colors in this table represent?

The color scheme was used to show the MIC in Table S10 of the original manuscript. In the revised manuscript, the colors have been removed ranges (**Extended Data Table 2**).

Reviewer #4 (Remarks to the Author):

This is an outstanding study that describes the discovery of a new class of penicillin-binding protein inhibitors designed to treat multidrug-resistant strains of *Neisseria gonorrhoea*, especially those strains that contain mosaic PBP2 mutations.

The design story describes fusing a boronic acid serine beta-lactamase warhead with Gram-negative cephalosporin sidechains to generate an initial PBP-targeted lead. This is followed by structure-guided rounds of lead optimization to ultimately generate VNRX-14079. It is an important antibiotic discovery story, well written and informative — just the sort of paper I would use in journal clubs for my trainees.

The study's high significance relates to the use of a non-beta-lactam scaffold to inhibit a PBP and, more specifically, generate potent compounds that can treat strains containing mosaic PBP2 mutations, which is no easy task. The lead's excellent drug-like properties include a low

propensity for side effects and reasonable PK properties. Microbiological studies show activity against a broad panel of *Neisseria gonorrhoea* isolates, including those strains that contain mosaic PBP2 mutations. These results suggest that VNRX-14079 is a viable drug candidate for advancing towards IND status.

The supplemental data is highly supportive and contains much valuable information. Supporting the rigor and reproducibility of the study and the advancement of VNRX-14079.

The conclusion section mentions the real-world financial limitations of developing novel antibacterial agents. However, in this case, the authors have added to this problem by developing a non-oral drug for *Neisseria gonorrhoea*, which limits the potential market size.

Recommendations

1. The PBP2 binding assays are performed on isolated/truncated enzymes. Providing inhibition data from unbiased PBP profiling on 21 using a standard bocillin-PBP binding assay (See PMID: 39739989) on Ng isolates (mosaic and non-mosaic) would add important supportive data to this manuscript.

We have added in vitro PBP binding data using the membranes prepared from ATCC 49226 (non-mosaic PBP2 strain) and H041 (mosaic PBP2 strain) in the revised manuscript (lines 266-273 and **Figure S3**).

2. Please provide information on the route of elimination and primary metabolism of 21

We have added data in the revised manuscript (lines 413-416, **Tables 3** and **S11**), showing that compound 21 has good solubility under physiological conditions (>700 μM) and is stable in both plasma and hepatocytes, implying that renal excretion is the main route of elimination. Further work, such as an animal PK study using an isotope-tagged compound, will need to be conducted for precise determination of the route of elimination; however, this is beyond the scope of this manuscript.

3. Information on the microsomal and plasma stability data of 21 should be included in Table 3. Microsomal and plasma stability data have been generated and added in the revised manuscript (**Tables 3**, **S11j**, and **S11k**).

4. Could the authors comment on the safety of 21 as a poly-anion and its propensity to chelate metals? How may this affect the distribution to infected tissues? And how does the distribution of 21 compare to CRO?

Compound **21** would not be expected to chelate metals, given that two acidic groups present (phosphonate and carboxylate) are not in close proximity to each other. Other clinically important beta-lactam antibiotics (e.g. cefixime, aztreonam) bear two acidic groups. The overall charge of **21** at pH 7.4 is anticipated to be -3. The mouse volume of distribution (0.334 L/kg) is higher than that of ceftriaxone (~0.1 L/kg), consistent with the relatively low plasma protein binding of Compound **21** (19% in mouse) vs. CRO (60% in mouse [PMID: 2291658]). Without detailed distribution studies, it's not possible further compare tissue distribution, but the in vivo efficacy study indicates that **21** is capable of reaching the target tissues in *N. gonorrhoeae* infection.

A point-by-point response to the reviewers' comments

Second external review: response to Reviewers' Comments (author response shown in purple)

Reviewer #1 (Remarks to the Author):

The authors have responded to my critiques satisfactorily. I have no further comments. I congratulate the authors on this fine piece of science. It is most unfortunate this program has been terminated because of a lack of interest from private funders. Such decisions will worsen the AMR crisis.

Thank you. It was totally unfortunate that the program was terminated due to the closure of the company discovery group.

Reviewer #2 (Remarks to the Author):

The authors have addressed my concerns in the revised manuscript. This is an impressive and exhaustive study.

Lisa Craig

Thank you for your positive feedback, Prof. Craig.

Reviewer #3 (Remarks to the Author):

In this revision, the authors have addressed my previous comments, including performing experiments on resistance emergence in WHO L and CDC-0197 strains after exposure to 4X, 16X, and sub-MIC concentrations of compound 21 and using isogenic mutants to explore the role of the porin and mtr efflux pump in resistance to compound 21.

I can also confirm that all the sequencing data associated with these experiments is available in NCBI's SRA. Since the mtr pump is the major antibiotic efflux pump in gonococcus, I was quite surprised by the premature stop codon in mtrE that was found in passage 8 of the WHO L experiment. I looked at the data myself and found the same mutations as the authors!

Thank you for your thorough review.

I have a couple of minor comments on the revision:

Line 349: Is the word efflux missing?

We have revised it from "the Mtr efflux" to "the Mtr efflux system".

Line 487-489: This blast search is mentioned for the first time in the discussion section. I would suggest putting this in the results section and mentioning this analysis in the methods as well.

This sentence has been moved to the Results section (lines 393-395). In addition, a sentence describing the BLAST analysis has been added to the Methods section (lines 539-542).

Extended Data Figure 7: It would be helpful if protein names used in the gonococcal literature were used here like in the main text (e.g. Opa proteins, MtrE).

Additional information about the protein names has been added to the figure legend to quickly identify the genes in which mutations were found.

Reviewer #4 (Remarks to the Author):

The authors have done an excellent job responding to the review, and I have no further suggestions for revisions. This is an outstanding antibacterial drug discovery paper that deserves publication.

Thank you for reviewing this manuscript.

Author's comment on revisions after expert review:

After the completion of expert reviews, the manuscript was significantly revised due to the editorial requirements for publication.

First external review: response to Reviewers' Comments (author response shown in purple)

Reviewer #1 (Remarks to the Author):

The authors have developed a series of novel benzoxaborinine-based penicillin-binding protein inhibitors (boro-PBPi) that bind to PBP2 and inhibit the growth of *N. gonorrhoeae*. Two compounds, 18 and 21, were tested for efficacy in the mouse vaginal colonization model using ovariectomized mice. Initially compound 18 at various doses was tested against lab strain FA1090 (sensitive to all antibiotics) to determine dosing guidelines. This was followed by testing compound 21 against CRO-resistant isolate H041 in the more commonly employed estradiol-treated mouse vaginal colonization model. The efficacy experiments in vivo have been clearly presented and the methods are robust. A couple of minor issues need clarification:

1. Can the authors clarify why initial testing done using ovariectomized mice and compound 18, while subsequent testing done in non-ovariectomized mice using compound 21?

The in vivo model with the susceptible FA1090 strain (used for compound **18**) was used as a proof of concept to show that boro-PBPi has in vivo efficacy against *N. gonorrhoeae*. In addition, practicality was another factor because this infection model was available at Eurofins through CARB-X's financial support and NIAID Pre-Clinical Services, while it was not possible to perform the in vivo study in Ann Jerse's lab in a timely manner due to operational challenges encountered during the COVID-19 pandemic. The in vivo model with the ceftriaxone-resistant H041 strain (used for compound **21**) was performed later by Ann Jerse's group to show the efficacy of boro-PBPi against this challenging strain. In the revised manuscript, we have added statements (lines 296-300 and 310-313) to clarify why the different infection models were used for compounds **18** and **21**.

2. In the efficacy experiment with compound 21, commensals (facultative aerobes) were monitored on HIA plates. Were there any obvious differences noted between the PBS groups and the antibiotic treated groups?

In the untreated and CRO-treated controls, as well as the test groups that received 10 mg/kg q12h, 200 mg/kg q24h, and 150 mg/kg q8h, 0-20% of mice had bacterial growth on HIA plates during the study. The majority of isolates were Gram-positive cocci in clusters, and some isolates were Gram-positive diplococci, probably *Staphylococcus* sp. and *Enterococcus* sp., respectively. In the test group (150 mg/kg, q12h), 60% of the mice were colonized with *Staphylococcus*-like bacteria. *Staphylococcal* presence in vaginal microbiota does not usually affect the duration of gonococcal colonization or the bioburden. Statements regarding this observation have been added in the revised Supplementary Information (added in the **Table S4** legend).

Reviewer #2 (Remarks to the Author):

The manuscript by Uehara et al. describes a novel class of benzoxaborinine-based PBP inhibitors (boro-PBPi) targeting drug resistant *Neisseria gonorrhoeae* including strains resistant

to ceftriaxone (CRO). They developed and tested a series of compounds containing the benzoxaborinine core used in novel beta-lactamase inhibitors, with cephalosporin side chains. As benzoxaborinines are not beta-lactams they are not susceptible to beta-lactamases and are thus not expected to contribute to their expansion as a resistance mechanism. They identified compound 21, a boro-PBPi bearing the N-methylsulfonyl-2-imidazolinone motif from mezlocillin, which binds with high affinity to PBP2, including some mosaic variants. Compound 21 has potent activity against CRO-resistant *N. gonorrhoeae* bearing these PBPs as well as an extended spectrum beta-lactamase-producing strain. Structural analysis of compound 21 precursors bound to PBP2 revealed a covalent interaction of the boron atom with Ser310. The b3-b4 loop is moved toward the active site, suggesting flexibility, which is associated with enhanced affinity. This compound has favorable pharmacokinetic properties and safety profile in mice. The efficacy of compound 21 against CRO-resistant *N. gonorrhoeae* was demonstrated in a murine infection model. The study concludes that compound 21 is a promising anti-gonorrhea agent poised for preclinical and clinical development, offering a potential new therapeutic option for multidrug resistant *N. gonorrhoeae*. The manuscript is well-written and well-illustrated, the study is thorough and well-executed and the results are compelling. Below are a list of issues that should be addressed, ordered as they appear in the manuscript.

Line 151 – “A series of phenol-bearing boro-PBPi was prepared ...” sounds like a phenol was added when in fact it is already present but was modified. Clarify and state the rationale for modifying the phenol.

A rationale has been added for clarification (lines 152-153).

Line 190 – Explain why a phosphonate is preferable to a phenoxide or carboxylate.

A statement has been added into the revised manuscript (lines 197-200) to clarify why the phosphonate substitution was examined. The superior activity of the phosphonate compound can be explained by the binding modes observed in the obtained PBP2/boro-PBPi complex structures (carboxylate boro-PBPi **12** vs. phosphonate **15**), which is described in the “Results” section “Structural evidence for high affinity binding of boro-PBPi to mosaic PBP2”.

Line 212-215 – The manuscript identifies compound 21 as the most efficacious, but presents structures of compounds 12 and 15 bound to PBP2. Were structures sought for compound 21 in complex with PBP2? Or mosaic PBP2? Compounds 12 and 15 differ from 21 in the side chain and phenol groups. Comment on where the side chains interact, how this might impact the b3-b4-loop and what these structures tell us about 21 binding. How might these boro-PBPis overcome the “conformational barrier created by resistance mutations”? (Line 236)

Since the original submission, we have now obtained a crystal structure of mosaic PBP2^{H041} in complex with 21. Its phosphonate forms similar interactions involving His514, Tyr543 and Arg503 as the phosphonate in **15**. Also similar to the complexes with **12** and **15**, the β_3 - β_4 loop occupies an inward position commensurate with higher-activity inhibitors of PBP2.

The new structure has been added into the revised manuscript and **Figure 2** accordingly.

It would be helpful for non-chemists to know exactly which motifs are used in which compound. In at least some of the structure figures (eg. those in S2) indicate the benzoxaborinine core and the cephalosporin-based side chains/moieties.

In **Figures S1 and S2**, the benzoxaborinine core and the β -lactam side chains have been color-labeled.

Explain why compound 18 was tested on CRO-susceptible FA1090 in the in vivo mouse model whereas 21 was tested with CRO-sensitive H041? Why wasn't 21 tested for both *N.*

gonorrhoeae strains? Why did the infection scheme differ for the two tests? What is the rationale for using ovariectomized BALB/c mice and what are the limitations? Why not use estradiol-treated mice with intact ovaries? (<https://pubmed.ncbi.nlm.nih.gov/31119637/>).

This in vivo model using susceptible FA1090 was used as a proof of concept to show that boron-PBPi has in vivo efficacy against *N. gonorrhoeae*. In addition, practicality was another factor because this infection model was available at Eurofins through CARB-X's financial support and NIAID Pre-Clinical Services, while it was not possible to perform the in vivo study in Ann Jerse's lab in a timely manner due to operational challenges encountered during the COVID-19 pandemic. In the revised manuscript, we have added statements to explain this (lines 296-300 and 310-313). The only murine infection model available at Eurofins was using ovariectomized BALB/c mice to establish colonization of *N. gonorrhoeae* FA1090.

Line 281-284: "Notably, neither the evolution of spontaneous gonococcal mutants nor a significant change in vaginal polymorphonuclear leukocytes (PMNs) was observed between groups during this study (Figure S10)." Reference Fig. S11 as well.

We appreciate your comments. **Fig. S11** of the original manuscript (**Extended Data Figure 6** in the revised manuscript) shows in vitro bactericidal activity not the in vivo study, so we did not add this figure as a reference, as suggested. Instead, we have added **Table S4 and S5** (line 328) as references, because these show the raw data of CFU and PMN measured in the in vivo study.

No *N. gonorrhoeae* mutants were identified upon treatment with 4x and 16x compound 21 and ceftriaxone but it is well known that ceftriaxone mutants arise in natural infections. The authors should test for mutant evolution using sub-inhibitory concentrations to identify mutations that might emerge during treatment. Whole genome sequencing can be used to identify resistance mutations/targets.

We performed 10-day passaging experiments with sub-MIC of compound **21** using ATCC 49226, WHO L, CDC-0197, and WHO Q strains. We obtained reduced susceptibility mutants from WHO L and CDC-0197 at day 8. These results have been added in the revised manuscript (lines 382-395 and **Extended Data Figure 7**).

Line 327 section – Provide context for each of the tests performed. Eg., define CYP enzymes and why these were selected for inhibition tests, why serine proteases, etc.

Clarification notes for these selectivity tests have been added in the revised manuscript (lines 402-410).

Line 341-391 – State here that compound 21 was administered subcutaneously in the in vivo efficacy test, in contrast to oral and intramuscular administration tested here.

We used subcutaneous administration in the in vivo efficacy studies because intramuscular administration in mice is challenging owing to the small injection volume allowed (0.1 mL/day) compared to the acceptable volume for SC administration (10 mL/kg/site/day) [reference PMID: 11180276]. We have added this information into the revised manuscript (lines 419-421).

The oral absorption of compound **21** (and other similar boro-PBPI) was expected to be poor in humans based on the Caco-2 data. Therefore, we did not use oral administration of boro-PBPI. Through our experience in discovery and development of the oral boron-based beta-lactamase inhibitor VNRX-7145 (i.e., ledaborbactam), we have found that ideal oral absorption of benzoxaborinine-containing compounds requires the use of prodrug forms of acidic moieties as well as low molecular weight for such prodrugs, which was challenging to accomplish for these boro-PBPI.

Was compound 21 tested on *N. gonorrhoeae* biofilms? On bacterial species other than *N. gonorrhoeae*?

We did not test the activity of **21** against *N. gonorrhoeae* biofilms. A study on biofilms is a future plan, as stated in the revised manuscript (line 496).

The activity of 21 against other bacterial species other than *N. gonorrhoeae* is shown in **Extended Data Table 1 (Table S1)** of the original manuscript).

The Discussion needs work. Several of the above points should be expanded upon in this section (structures, mouse models, resistance) and the authors should outline next steps to bring this compound to clinical tests. This section ends rather abruptly and suggests that 21 will not be advanced further.

As suggested, the Discussion section of the revised manuscript has been expanded on crystal structures, mouse models, resistance, and future steps. Additionally, we have revised the last paragraph of Discussion (lines 491-500), replacing the previous commentary on the current real-world situation of the antibiotic industry with next steps to advance compound **21** as an anti-gonorrhea agent. However, it is pertinent to highlight the current challenging business environment surrounding antibiotic discovery and development. For instance, in mid-2025, Venatorx terminated all discovery and development activities, resulting in the layoff of nearly all employees, including all Venatorx authors of this manuscript. The discovery of **21** failed to attract interest from private funding sources, such as venture capital, primarily due to the lack of anticipated profitability in developing an anti-gonorrhoeae drug within the current broken antibiotic market.

Reviewer #3 (Remarks to the Author):

In this manuscript by Uehara and colleagues, the development of a novel class of penicillin binding protein (PBP) inhibitors is described. Additionally, the authors demonstrate activity against *Neisseria gonorrhoeae*, including against strains with reduced susceptibility to the current recommended therapy ceftriaxone both in vitro and in the mouse model for gonorrhea.

As *N. gonorrhoeae* has developed resistance to all previous recommended therapies, development of new antimicrobials targeting *N. gonorrhoeae* is of utmost importance. The manuscript is well written, and I found it enjoyable to read! As my own expertise is related to the genetic mechanisms underlying antimicrobial resistance in *N. gonorrhoeae*, my major comments on the manuscript are related to the potential for emergence of resistance to these novel compounds:

1. The authors found that among a panel of strains, two encoding mutations at codon 501 in *penA* had elevated MICs to compound 21. The authors suggest that resistance associated with these mutations are not of major concern because A501P has a demonstrated fitness cost and strains encoding this mutation are no longer transmitting. It may be worth mentioning here that while A501P is not commonly observed, A501V is present in currently circulating lineages (e.g. <https://www.microbiologyresearch.org/content/journal/mgen/10.1099/mgen.0.000480>).

A statement describing the presence of A501V in currently circulating lineages has been added in the revised manuscript, as suggested (lines 345-347).

2. Likewise, the authors demonstrated a low frequency of resistance for three *N. gonorrhoeae* strains, including WHO Q encoding *penA* 60. Do the authors expect that strains already encoding A501V or strains encoding *penA* 34 (where the A501P previously emerged) are more likely to evolve resistance to the novel compounds developed here?

We performed FoR experiments WHO L and CDC-0197(*penA*34). The results have been added into the revised manuscript (lines 377-380 and **Table S10**).

Additional minor comments:

The authors use a strain panel including strains that are part of the WHO panel. However, some strains are referred to with WHO nomenclature and others with the original strain names. For example, F89 is WHO Y, and H041 is WHO X. Since WHO nomenclature is used for other strains in the study, I would suggest using WHO nomenclature throughout.

The names H041 and F89 have been used in many publications, whereas the corresponding WHO strain names (WHO X and Y, respectively) have not. Other WHO strains do not have general strain names other than the WHO names. Thus, we retained the strain names as they are. However, in the revised manuscript, the WHO strain designations have been incorporated at the initial mention of H041 and F89 (lines 135 and 337, respectively).

Line 135: As there are several mosaic PBP2 alleles that mediate reduced susceptibility to ceftriaxone, it would be helpful to list the *penA* allele encoded by H041/WHO X; particularly as *penA* 60 is mentioned in the introduction, and H041 does not encode *penA* 60. Likewise, it would be helpful to report the relevant *penA* alleles on Lines 206 and 215.

The corresponding *penA* alleles have been added in the revised manuscript (lines 135-136).

Line 148-149: Given the PBP2 binding data in Figure 1, it seems clear that variation associated with mosaic *penA* alleles contributes to activity of these compounds, but I wonder if the authors have considered whether mutation in *porB* (encoding the porin) or the *mtr* operon (encoding and

efflux pump) are also contributing to MICs of the novel BLIs studied in this manuscript. FA19 and FA1090 are also WT at those loci, which contribute to resistance to multiple drugs. In the revised manuscript, we have added MIC data against isogenic strains to address the effect of the PorB porin and Mtr efflux system on boro-PBPi (**Table S8**). Minimal effect of the PorB porin and the Mtr efflux was observed on boro-PBPi activity (lines 348-350 in the revised manuscript).

Line 314/Table 2: This CDC strain panel is not particularly representative of the *N. gonorrhoeae* population as the majority of strains come from a single *penA* 34 encoding lineage of *N. gonorrhoeae*.

The CDC strain panel was formed from surveillance studies conducted in the US, not based on genetic diversity. The *penA* information is not described in the main text reference 61 or a public information. Our genome sequencing analysis revealed for the first time that most of these CDC strains carried *penA*-34. The sentence has been modified to clarify the points in the revised manuscript (lines 366-368).

The activity against the WHO reference strain panel listed in **Table 1** reflects the activity of boron-based PBP inhibitors against more diverse strains than that of the CDC strain panel.

Line 324-325: It seems to me that it would be interesting to know the frequency of resistance for *penA* 34 encoding strains since 1) this lineage is still transmitting and 2) this lineage is only a single mutation from encoding *penA* 42.

As described above, FoR studies with WHO L and CDC-0197 were performed, and the data have been added in **Table S10** (4x MIC and 16x MIC) in the revised manuscript (lines 377-380). We did not obtain any colonies from either strain, and the FoR was $<5 \times 10^{-9}$ against these strains.

Line 491: Have reads been deposited in a public repository?

WGS reads were deposited in the NCBI GenBank with the BioProject ID: PRJNA1353147.

Figure 1: There are a lot of colors associated with the table in this figure, and they are not described in the figure legend. In particular, the red/green color scheme for low/high PBP2 binding and MICs may not be ideal for readers with colorblindness.

As suggested, the table in **Figure 1** has been updated in the revised manuscript.

Figure 3: It appears that the legend for panels B and C are mixed up.

Thank you. The panels B and C have been swapped to align with the legend.

Table S7: Is this genotyping from previously reported results? If not, are the methods used to identify variants reported in the methods?

The genotypes of these strains have been previously described (references 73 and 74 in the main text). To confirm that the strains used were correct, we performed and analyzed genome sequencing. The description of the methods used for the WGS analysis and **Table S13** have

been added in the revised manuscript, and the raw sequence data have been deposited in the GenBank database (PRJNA1353147).

Table S10: What do the colors in this table represent?

The color scheme was used to show the MIC in Table S10 of the original manuscript. In the revised manuscript, the colors have been removed ranges (**Extended Data Table 2**).

Reviewer #4 (Remarks to the Author):

This is an outstanding study that describes the discovery of a new class of penicillin-binding protein inhibitors designed to treat multidrug-resistant strains of *Neisseria gonorrhoea*, especially those strains that contain mosaic PBP2 mutations.

The design story describes fusing a boronic acid serine beta-lactamase warhead with Gram-negative cephalosporin sidechains to generate an initial PBP-targeted lead. This is followed by structure-guided rounds of lead optimization to ultimately generate VNRX-14079. It is an important antibiotic discovery story, well written and informative — just the sort of paper I would use in journal clubs for my trainees.

The study's high significance relates to the use of a non-beta-lactam scaffold to inhibit a PBP and, more specifically, generate potent compounds that can treat strains containing mosaic PBP2 mutations, which is no easy task. The lead's excellent drug-like properties include a low propensity for side effects and reasonable PK properties. Microbiological studies show activity against a broad panel of *Neisseria gonorrhoea* isolates, including those strains that contain mosaic PBP2 mutations. These results suggest that VNRX-14079 is a viable drug candidate for advancing towards IND status.

The supplemental data is highly supportive and contains much valuable information. Supporting the rigor and reproducibility of the study and the advancement of VNRX-14079.

The conclusion section mentions the real-world financial limitations of developing novel antibacterial agents. However, in this case, the authors have added to this problem by developing a non-oral drug for *Neisseria gonorrhoea*, which limits the potential market size.

Recommendations

1. The PBP2 binding assays are performed on isolated/truncated enzymes. Providing inhibition data from unbiased PBP profiling on 21 using a standard bocillin-PBP binding assay (See PMID: 39739989) on Ng isolates (mosaic and non-mosaic) would add important supportive data to this manuscript.

We have added in vitro PBP binding data using the membranes prepared from ATCC 49226 (non-mosaic PBP2 strain) and H041 (mosaic PBP2 strain) in the revised manuscript (lines 266-273 and **Figure S3**).

2. Please provide information on the route of elimination and primary metabolism of 21

We have added data in the revised manuscript (lines 413-416, **Tables 3** and **S11**), showing that compound 21 has good solubility under physiological conditions ($>700 \mu\text{M}$) and is stable in both plasma and hepatocytes, implying that renal excretion is the main route of elimination. Further work, such as an animal PK study using an isotope-tagged compound, will need to be conducted for precise determination of the route of elimination; however, this is beyond the scope of this manuscript.

3. Information on the microsomal and plasma stability data of 21 should be included in Table 3.

Microsomal and plasma stability data have been generated and added in the revised manuscript (**Tables 3, S11j, and S11k**).

4. Could the authors comment on the safety of 21 as a poly-anion and its propensity to chelate metals? How may this affect the distribution to infected tissues? And how does the distribution of 21 compare to CRO?

Compound **21** would not be expected to chelate metals, given that two acidic groups present (phosphonate and carboxylate) are not in close proximity to each other. Other clinically important beta-lactam antibiotics (e.g. cefixime, aztreonam) bear two acidic groups. The overall charge of **21** at pH 7.4 is anticipated to be -3. The mouse volume of distribution (0.334 L/kg) is higher than that of ceftriaxone (~ 0.1 L/kg), consistent with the relatively low plasma protein binding of Compound **21** (19% in mouse) vs. CRO (60% in mouse [PMID: 2291658]). Without detailed distribution studies, it's not possible further compare tissue distribution, but the in vivo efficacy study indicates that **21** is capable of reaching the target tissues in *N. gonorrhoeae* infection.